# KAT6B is required for histone 3 lysine 9 acetylation and SOX gene expression in the developing brain

Maria I Bergamasco[1,2], Waruni Abeysekera[1,2], Alexandra L Garnham[1,2], Yifang Hu[1,2], Connie SN Li-Wai-Suen[1,2], Bilal N Sheikh[1,2], Gordon K Smyth[1,3], Tim Thomas[1,2,*], Anne K Voss[1,2,*]

**Heterozygous mutations in the histone lysine acetyltransferase gene _KAT6B_ (_MYST4/MORF/QKF_) underlie neurodevelopmental disorders, but the mechanistic roles of KAT6B remain poorly understood. Here, we show that loss of KAT6B in embryonic neural stem and progenitor cells (NSPCs) impaired cell proliferation, neuronal differentiation, and neurite outgrowth. Mechanistically, loss of KAT6B resulted in reduced acetylation at histone H3 lysine 9 and reduced expression of key nervous system development genes in NSPCs and the developing cortex, including the SOX gene family, in particular _Sox2_, which is a key driver of neural progenitor proliferation, multipotency and brain development. In the fetal cortex, KAT6B occupied the _Sox2_ locus. Loss of KAT6B caused a reduction in _Sox2_ promoter activity in NSPCs. _Sox2_ overexpression partially rescued the proliferative defect of _Kat6b_$^{-/-}$ NSPCs. Collectively, these results elucidate molecular requirements for KAT6B in brain development and identify key KAT6B targets in neural precursor cells and the developing brain.**

## Introduction

Genetic mutations in epigenetic regulators commonly underly intellectual disability disorders (Kleefstra et al, 2014; Kochinke et al, 2016). One of the most abundant epigenetic modifications is histone lysine acetylation, catalyzed by lysine acetyltransferases (KATs) and associated with transcriptional activation. Heterozygous mutations in the MYST family histone lysine acetyltransferase gene, _KAT6B_ (_MYST4/QKF/MORF_) cause the Say-Barber-Biesecker-Young-Simpson variant of Ohdo syndrome, Genitopatellar syndrome, and similar disorders (Clayton-Smith et al, 2011; Campeau et al, 2012; Simpson et al, 2012), defined by a global development delay and cognitive impairment.

The _Kat6b_ mRNA expression pattern has been extensively studied throughout development in mice. Unlike many chromatin regulators, _Kat6b_ is subject to significant regulation at the mRNA level. _Kat6b_ mRNA levels are low at embryonic day 9.5 (E9.5) but become up-regulated from E10.5, with substantial increases observed during cerebral cortex development from E11.5 and peaking in the E15.5 cortex (Thomas et al, 2000). Despite declining during neuronal differentiation, mRNA levels remain elevated in the subventricular neurogenic proliferation zone at E17.5, as well as on postnatal days 0, 1, 7, 14, 21, and in adulthood (Thomas et al, 2000; Merson et al, 2006). Congruently, _Kat6b_ promoter activity is high during adult neurogenesis in the neural stem cell (NSC) population and gradually decreases as cells differentiate (Sheikh et al, 2012). Beyond the developing and adult brain, _Kat6b_ mRNA levels are also elevated in developing facial structures, including the E11.5 and E12.5 frontal nasal and maxillary processes, the E15.5 eyelids, and in tooth primordia (Thomas et al, 2000; Clayton-Smith et al, 2011; Kraft et al, 2011). Elevated _Kat6b_ mRNA or promoter activity were also observed in the anterior aspects of the developing limb bud at E10.5 and E11.5, limb skeletal elements at E12.5, E14.5, and E15.5 (Clayton-Smith et al, 2011; Kraft et al, 2011) and in rib/intercostal structures (Kraft et al, 2011).

Consistent with strong _Kat6b_ expression in the developing cerebral cortex, embryos, fetuses and mice deficient in _Kat6b_ mRNA displayed defects in cortex development. A reduction in the number of proliferating cells in the dorsal telencephalon at E11.5 and reduced numbers of differentiating cells in the developing fetal cerebral cortex were observed, but the rate of cell death was not affected by KAT6B status (Thomas et al, 2000). Consequently, _Kat6b_ deficient mice have a small cortical plate during development and a small cerebral cortex in adulthood with fewer _Otx1_-positive large cortical pyramidal neurons in cortical layer V and fewer GAD67-positive interneurons in the cortex (Thomas et al, 2000), as well as fewer NSCs in the subventricular zone, a reduced number of migrating neuroblasts in the rostral migratory stream and smaller olfactory bulbs (Merson et al, 2006). Furthermore, _Kat6b_ deficient NSCs isolated from the adult mouse brain display impaired self-renewal and neuronal differentiation (Merson et al, 2006). Despite this clear requirement for KAT6B in the developing and adult brain

---

[1]The Walter and Eliza Hall Institute of Medical Research, Parkville, Australia   [2]Department of Medical Biology, The University of Melbourne, Parkville, Australia   [3]School of Mathematics and Statistics, University of Melbourne, Parkville, Australia

Correspondence: avoss@wehi.edu.au; tthomas@wehi.edu.au
*Tim Thomas and Anne K Voss contributed equally to this work and share senior authorship

and in NSCs, how KAT6B controls neural development and progenitor activity remains poorly understood at the molecular level.

NSCs are multipotent cells that can be isolated from the developing and adult brain. In vitro, these cells form a heterogeneous population comprising NSCs and partially or completely lineage restricted progenitor cells, collectively referred to as neural stem and progenitor cells (NSPCs). NSCs and neural precursor cells give rise to the cellular diversity of the developing and adult nervous system (reviewed in Kriegstein and Alvarez-Buylla [2009]). NSCs must sustain adequate self-renewing divisions whilst also committing to neuronal or glial lineages as required. During differentiation, NSCs undergo extensive changes in gene expression, activating lineage-specific gene expression programs and silencing stem cell-associated genes. Various epigenetic mechanisms have been documented to control these processes (Yoon et al, 2018). For example, the histone methyltransferases EZH2, EHMT1, EHMT2, and DOT1L inhibit premature NSPC differentiation in a cell-based model (Ciceri et al, 2024). More specifically, DOTL1 restricts neural progenitor differentiation by ensuring the access of the stem cell transcription factor SOX2 to its target genes, thereby promoting the stem cell transcription program (Ferrari et al, 2020). The histone methyltransferase MLL1 maintains positional information of NSCs (Delgado et al, 2020), and the histone acetyltransferase KAT7 is essential for de novo gene activation during neuronal differentiation (Kueh et al, 2023). These examples underpin the importance of epigenetic control in brain development and NSCs function.

Despite the well-documented importance of KAT6B in brain development, the molecular function of KAT6B in neural cell types, including histone acetylation and gene targets, have not been reported. Using $Kat6b$ loss and gain of function mice, as well as epigenomic and transcriptomic profiling, we report here that KAT6B controls neural precursor cells through the activation of key developmental control genes, including $Sox2$.

# Results

### KAT6B is essential for histone H3 lysine 9 acetylation

To investigate molecular targets of KAT6B during mouse development, we used $Kat6b$ mutant mice ($Kat6b^-$ [Bergamasco et al, 2024a, 2024b]), which lack exons 2–12 of the endogenous $Kat6b$ gene, and $Kat6b$ BAC transgenic mice ($Tg(Kat6b)$ [Bergamasco et al, 2024a]) which overexpress $Kat6b$ ~4.5-fold above endogenous levels. The presence of the $Tg(Kat6b)$ transgene rescues the hematopoietic defects of $Kat6b^{-/-}$ mice (Bergamasco et al, 2024a), showing that the $Tg(Kat6b)$ transgene produces functional KAT6B protein.

At embryonic day 12.5 (E12.5), a stage at which KAT6B is highly expressed (Thomas et al, 2000), $Kat6b^{+/-}$ and $Kat6b^{-/-}$ mutant embryos were externally indistinguishable from wild-type controls ($Kat6b^{+/+}$; Fig 1A). Both genotypes were observed at expected Mendelian ratios in utero; however, $Kat6b^{-/-}$ mice were not present at weaning (3 wk of age; $P < 10^{-6}$; Fig S1A) and $Kat6b^{+/-}$ animals were 18% underrepresented at weaning compared with controls ($P = 5 \times 10^{-6}$; Fig S1B), indicating that haploinsufficiency for $Kat6b$ impairs survival in early life.

Compared with WT controls, $Kat6b$ heterozygous ($Kat6b^{+/-}$) and null ($Kat6b^{-/-}$) samples displayed a 38–48% and 100% reduction in $Kat6b$ mRNA, respectively, in E12.5 dorsal telencephalon and NSPCs ($P < 10^{-6}$ to 0.001; Figs 1B and C and S1C and D). Conversely, the E12.5 dorsal telencephalon of $Tg(Kat6b)$ mice overexpressed $Kat6b$ approximately fourfold above endogenous levels (Fig S1E).

To investigate the histone lysine targets of KAT6B during mouse development, we assessed histone acetylation at seven lysines on histone H3 and four lysines on histone H4 by western immunoblotting in whole E12.5 embryos either lacking $Kat6b$ ($Kat6b^{-/-}$) or overexpressing $Kat6b$ [$Tg(Kat6b)$]. Of the 11 histone residues examined, H3K9 was the only histone residue at which acetylation levels were reduced in $Kat6b^{-/-}$ versus control embryos (47% reduction, $P = 8 \times 10^{-6}$) and increased in $Tg(Kat6b)$ versus control embryos (43% increase, $P = 0.03$; Fig S2A and B), as would be expected if KAT6B acetylated this residue. H3K14ac was increased 1.7-fold in $Kat6b^{-/-}$ versus control embryos ($P = 2 \times 10^{-6}$) but was unaffected in $Tg(Kat6b)$ embryos. Acetylation levels at the other nine histone H3 and H4 lysine residues (H3K4, H3K18, H3K23, H3K27, H3K56, H4K5, H4K8, H4K12, H4K16) were not significantly affected by loss or gain of KAT6B (Fig S2A and B).

Congruent with the findings in whole E12.5 embryos, we found a 36% and 12% reduction in H3K9ac in $Kat6b^{-/-}$ NSPCs and E12.5 dorsal telencephalon, respectively, compared with control samples ($P = 0.003$ and 0.01; Fig 1D–G). In addition, H3K23ac was reduced by 21% in the $Kat6b^{-/-}$ dorsal telencephalon compared with control samples ($P = 0.02$), but not significantly changed in $Kat6b^{-/-}$ NSPCs (Fig 1H–K). Our data suggest a role for KAT6B in H3K9 acetylation and, in addition, a possible role in H3K23 acetylation in some tissues.

To investigate the locus-specific effects of KAT6B, we assessed H3K9ac, H3K14ac, and H3K23ac levels and RNA polymerase II, subunit A (POLR2A) occupancy by CUT&Tag and DNA accessibility by ATAC-sequencing in $Kat6b^{+/+}$ and $Kat6b^{-/-}$ NSPCs. In WT samples, H3K9, H3K14 and H3K23 acetylation levels correlated strongly with POLR2A occupancy at all protein coding genes ($R^2 = 0.5$–0.6; Fig S3A) and at brain development genes (GO:0007420~brain development; $R^2 = 0.5$–0.7; Fig S3B).

H3K9ac was reduced at the transcription start site (TSS ± 1 kb) in $Kat6b^{-/-}$ versus $Kat6b^{+/+}$ NSPCs ($P < 10^{-6}$; Fig 1L). Significant loss of H3K9ac was observed individually at 112 promoters, 3,452 gene bodies and 44 active enhancers (false discovery rate [FDR] < 0.05; Figs 1M–O and S3C; Table S1) and collectively at promoters, gene bodies, and active enhancers ($P < 10^{-6}$; Fig 1P). While loss of KAT6B did not appear to cause genome-wide effects on H3K23ac in NSPCs (Fig 1H and I), we investigated a locus-specific effect. H3K23ac coverage was reduced at the TSS (±1 kb) in $Kat6b^{-/-}$ versus $Kat6b^{+/+}$ NSPCs ($P < 10^{-6}$; Figs 1Q and S3D; Table S1) and was collectively slightly reduced at promoters, gene bodies, and active enhancers ($P < 10^{-6}$; Fig 1R). POLR2A coverage over the TSS ± 1 kb and at active enhancers was reduced in $Kat6b^{-/-}$ versus $Kat6b^{+/+}$ NSPCs ($P < 10^{-6}$; Figs 1S and T and S3E). Significant loss of POLR2A occupancy was observed at 48 individual enhancers (Fig S3F; Table S1).

Similar to the unexpected increase in H3K14ac in whole $Kat6b^{-/-}$ versus $Kat6b^{+/+}$ E12.5 embryos (Fig S2A and B), we also observed an increase in H3K14ac by Western blotting in $Kat6b^{-/-}$ versus $Kat6b^{+/+}$ NSPCs ($P = 0.01$), but not in the dorsal telencephalon (Fig S4A–D). Assessed by CUT&Tag, H3K14ac was elevated in $Kat6b^{-/-}$ versus

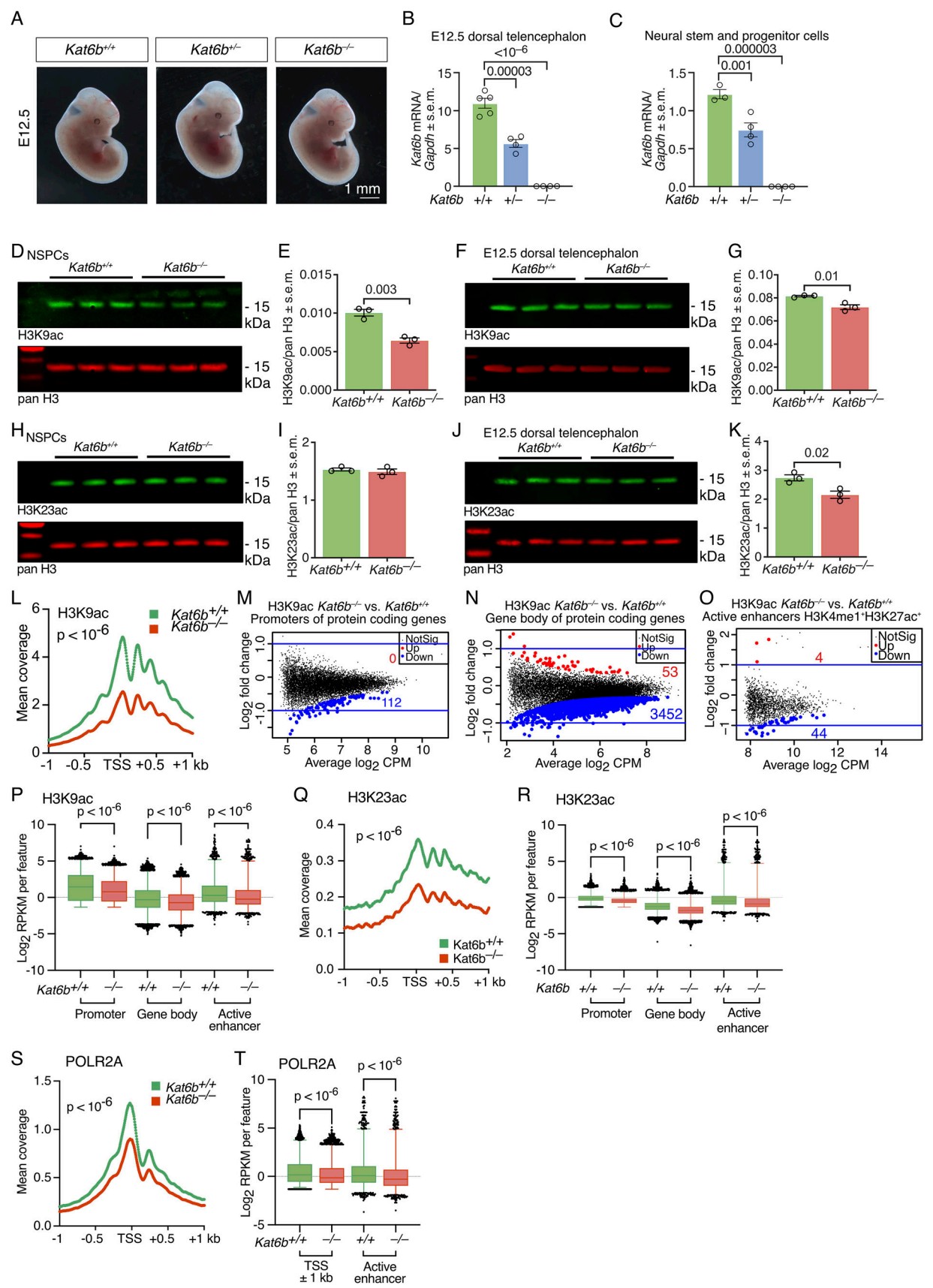

$Kat6b^{+/+}$ NSPCs at 12,180 promoters, 15,943 gene bodies and 5,193 active enhancers (Fig S4E–L; Table S1). Similarly, DNA accessibility assessed by ATAC-sequencing was increased in $Kat6b^{-/-}$ versus $Kat6b^{+/+}$ NSPCs at 14,000 promoters ($P < 10^{-6}$; Fig S4M–O; Table S1).

Taken together, our data suggest that KAT6B is essential for normal H3K9 acetylation levels at a large number of genes. In contrast, the requirements for H3K23ac were more restricted to a smaller number of genes. Curiously, we found that the presence of KAT6B limited H3K14ac. Our peculiar observations of increased H3K14ac and DNA accessibility in $Kat6b^{-/-}$ versus $Kat6b^{+/+}$ may suggest increased activity of another histone acetyltransferase.

### Loss of KAT6B caused a reduction in expression of genes required for brain development

KAT6B is highly expressed in the neurogenic subventricular zone and NSCs (Merson et al, 2006; Sheikh et al, 2012). To assess the role of KAT6B in gene expression in NSPCs, we performed RNA-sequencing. RNA-sequencing profiles of $Kat6b^{-/-}$ versus $Kat6b^{+/+}$ NSPCs segregated by genotype (Fig 2A). $Kat6b$ deletion resulted in significant down-regulation of 3,861 genes and up-regulation of 3,833 genes, while $Kat6b$ overexpression [$Tg(Kat6b)$] caused the up-regulation of 2,322 genes and the down-regulation of 2,561 genes (FDR< 0.05; Figs 2B and C and S5A–C; Table S2). Genes down-regulated in $Kat6b^{-/-}$ versus $Kat6b^{+/+}$ NSPCs were generally up-regulated in $Tg(Kat6b)$ NSPCs ($P = 0.0001$; Fig 2D), supporting a role of KAT6B in the direct or indirect regulation of these genes. Genes down-regulated in $Kat6b^{-/-}$ versus $Kat6b^{+/+}$ NSPCs were enriched for nervous system development, gene transcription and metabolic processes, whereas up-regulated genes lacked such annotations (Figs 2E and S5D, Table S2). A subset of genes essential for NSC function was down-regulated (Fig S5E). The top 20 brain development genes (GO:0007420) down-regulated at the RNA level displayed a reduction in both H3K9ac and mRNA levels (Fig 2F; Tables S1 and S2) and included early regulators of neuroepithelium, brain development, and NSC function, e.g., $Sox1$, $Sox2$, $Pax6$, and $Id2$ (Schmahl et al, 1993; Malas et al, 2003; Ferri et al, 2004; Niola et al, 2012), as well as regulators of neurite elongation, $Sall1$, $Hap1$, and $Rac3$ (Orioli et al, 2006; Rong et al, 2006; Harrison et al, 2008),

identifying these as putative KAT6B target genes. Curiously, these same down-regulated genes had an increase in H3K14ac and DNA accessibility (Fig S5F). Neural precursor proliferation genes (GO: 0061351) were profoundly affected by loss of KAT6B (Fig 2G). At brain development genes and neural precursor proliferation genes, we observed a positive correlation between H3K9ac and POLR2A read counts per gene (log2 count per million [CPM]; Fig 2H and I). Correlations with RNA levels were strongest when the size of the gene was taken into consideration (log2 RPKM; Fig 2J–M). Positive correlations were observed between H3K9ac and mRNA levels ($R^2 = 0.6$ and 0.5, respectively; both $P < 10^{-6}$) and between H3K9ac and POLR2A levels (both $R^2 = 0.7$ and $P < 10^{-6}$; Fig 2H–M). Notably, the fold-changes in H3K9ac and mRNA levels due to $Kat6b$ deletion also correlated significantly in brain development genes ($R^2 = 0.2$; $P = 0.002$), and neural precursor proliferation genes ($R^2 = 0.3$; $P = 0.004$; Fig 2N and O).

We explored a possible compensatory regulation of other histone acetyltransferase or deacetylases. We postulated that any compensatory mechanism would operate in opposite directions in $Kat6b^{-/-}$ versus $Kat6b^{+/+}$ and $Tg(Kat6b)$ versus $Kat6b^{+/+}$ NSPCs. Of the eight genes that, apart from $Kat6b$, encode nuclear histone acetyltransferases with defined acetyl-co-enzyme A binding sites, only $Kat2a$ ($Gcn5$) mRNA was statistically significantly changed, namely up-regulated in $Kat6b^{-/-}$ versus $Kat6b^{+/+}$ and down-regulated in $Tg(Kat6b)$ versus $Kat6b^{+/+}$ NSPCs (Fig S5G). In addition, two histone deacetylase genes, $Hdac11$ and $Sirt2$ were down-regulated in $Kat6b^{-/-}$ versus $Kat6b^{+/+}$ and up-regulated in $Tg(Kat6b)$ versus $Kat6b^{+/+}$ NSPCs (Fig S5G).

### Loss of KAT6B caused down-regulation of brain development genes in vivo

Examining the effects of loss and gain of KAT6B on the developing cortex, we found 64 genes were differentially expressed in $Kat6b^{-/-}$ versus $Kat6b^{+/+}$ E12.5 dorsal telencephalon (FDR < 0.05; 50 down-regulated and 14 up-regulated) and 1,405 genes were differentially expressed in $Tg(Kat6b)$ versus $Kat6b^{+/+}$ E12.5 dorsal telencephalon (779 up-regulated and 626 down-regulated; Figs 3A–C and S5H; Table S3). Genes down-regulated in $Kat6b^{-/-}$ versus $Kat6b^{+/+}$ or up-

**Figure 1. Effects of *Kat6b* deletion on histone acetylation and RNA polymerase II occupancy.**
**(A)** Representative images of E12.5 $Kat6b^{+/+}$, $Kat6b^{+/-}$ and $Kat6b^{-/-}$ embryos. Scale bar 1 mm. **(B, C)** RT-qPCR detecting $Kat6b$ mRNA levels in $Kat6b^{+/+}$, $Kat6b^{+/-}$, and $Kat6b^{-/-}$ E12.5 dorsal telencephalon (B) and cultured neural stem and progenitor cells (NSPCs) (C), normalized to $Gapdh$ mRNA. **(D, E, F, G, H, I, J, K)** Fluorescent Western immunoblots and quantification of H3K9ac (D, E, F, G) and H3K23ac (H, I, J, K) with pan H3 loading control in passage 4–5 $Kat6b^{+/+}$ versus $Kat6b^{-/-}$ E12.5 NSPCs (D, E, H, I) or E12.5 dorsal telencephalon (F, G, J, K). Each lane (D, F, H, J) contains protein from an individual embryo or NSPCs isolated from an individual embryo. Each circle (E, G, I, K) represents one lane of the associated immunoblot. 500 ng (D, F) and 250 ng (H, J) of acid extracted protein were loaded per lane. **(L, M, N, O, P, Q, R, S)** CUT&Tag sequencing results detecting the genomic distribution of H3K9ac, H3K23ac and RNA polymerase II, subunit A (POLR2A) in passage 5 $Kat6b^{-/-}$ versus $Kat6b^{+/+}$ E12.5 NSPCs. N = 4 embryos per genotype. CUT&Tag data were analyzed as described in the methods section. FDR < 0.05 was considered significant. **(L)** Coverage plot of H3K9ac reads across the transcription start site ± 1 kb in $Kat6b^{+/+}$ and $Kat6b^{-/-}$ NSPCs. **(M, N, O)** Log2-fold change versus average log2 read count (CPM) of H3K9ac in $Kat6b^{-/-}$ versus $Kat6b^{+/+}$ NSPCs at the promoters (M) and gene bodies of protein coding genes (N) or active enhancers (O) defined as H3K4me1$^+$H3K27ac$^+$ in GSM2406793 and GSM2406791 (Bertolini et al, 2019). Regions with significantly reduced H3K9ac shown in blue and increased H3K9ac in red. **(P)** H3K9ac read counts per kilobase per one million reads (RPKM) at promoters and gene bodies of all protein coding genes and active enhancers in $Kat6b^{+/+}$ and $Kat6b^{-/-}$ NSPCs. **(Q)** Coverage plot of H3K23ac reads across the transcription start site ± 1 kb in $Kat6b^{-/-}$ versus $Kat6b^{+/+}$ NSPCs. **(R)** H3K23ac read counts per kilobase per one million reads (RPKM) at promoters and gene bodies of all protein coding genes and active enhancers in $Kat6b^{+/+}$ and $Kat6b^{-/-}$ NSPCs. **(S)** Coverage plot of POLR2A reads across the transcription start site ± 1 kb in $Kat6b^{-/-}$ versus $Kat6b^{+/+}$ NSPCs. **(T)** POLR2A read counts across the transcription start site ± 1 kb of all protein coding genes and active enhancers in $Kat6b^{+/+}$ and $Kat6b^{-/-}$ NSPCs. N = NSPC cultures or E12.5 dorsal telencephalon samples from three to five embryos per genotype (A, B, C, D, E, F, G, H, I, J, K) and from four embryos per genotype (L, M, N, O, P, Q, R, S, T). (B, C, E, G, I, K) Each circle represents tissue or cells isolated from an individual embryo (B, C, E, G, I, K). Data are displayed as mean ± SEM, mean (L, Q, S), log2 of the fold-change (M, N, O) or 1–99 percentile plus outliers (P, R, T). Data were analyzed using a one-way ANOVA with Dunnett correction (B, C), an unpaired $t$ test (E, G, I, K), as described in the methods section (L, M, N, O, Q, S) or by Kruskal-Wallis test (P, R, T).
Source data are available online for this figure.

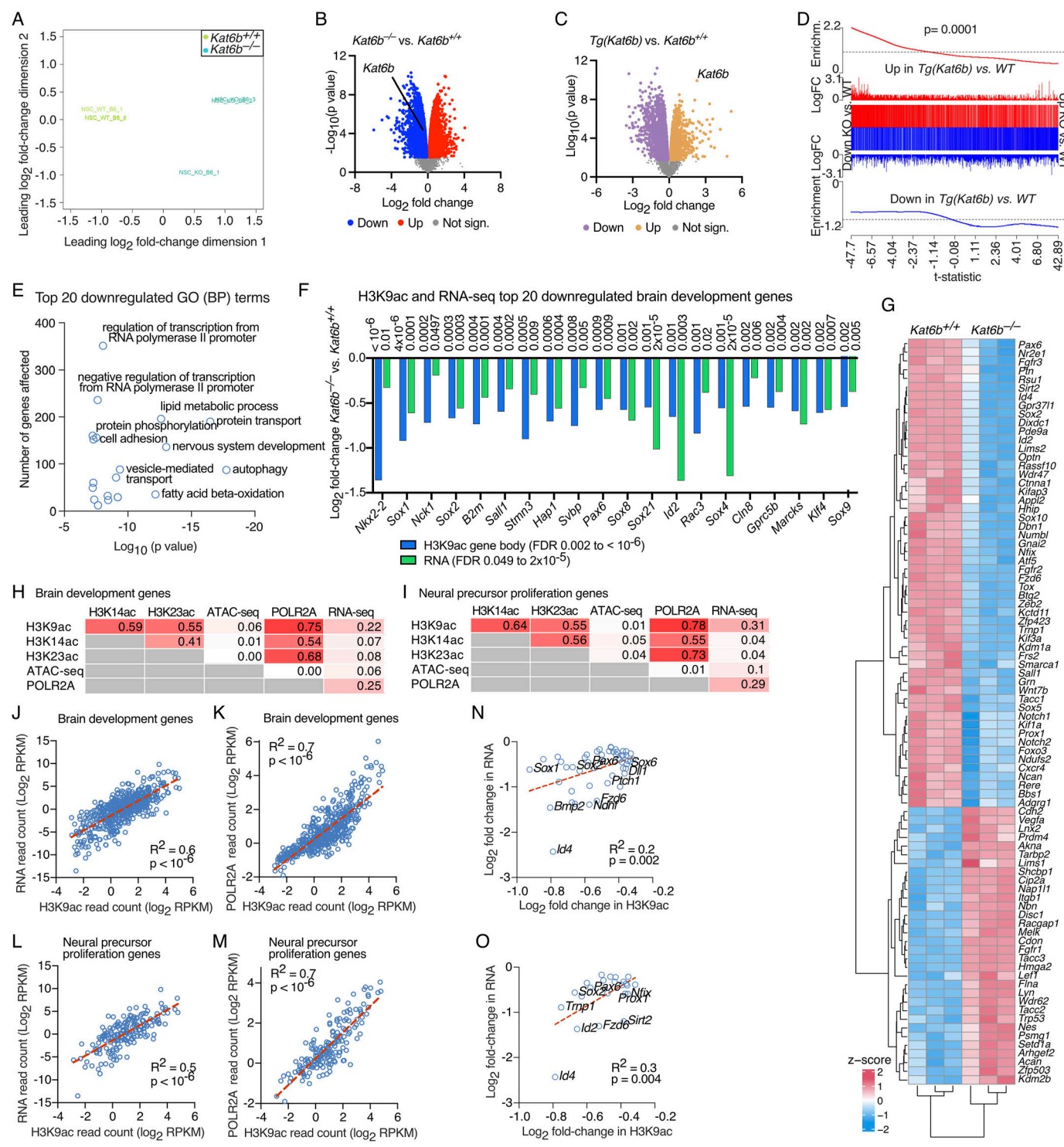

**Figure 2. Effects of loss of KAT6B on gene expression in NSPCs.**
**(A, B, C, D, E, G)** RNA-sequencing data of N = 3 *Kat6b*$^{-/-}$ versus 3 *Kat6b*$^{+/+}$ passage 5 NSPC isolates and 6 *Tg(Kat6b)* versus 2 *Kat6b*$^{+/+}$ NSPC isolates. **(F, H, I, J, K, L, M, N, O)** Comparison of RNA-sequencing results to CUT&Tag and ATAC-seq results. For CUT&Tag and ATAC-seq experiments, N = 4 embryos per genotype. The CUT&Tag, ATAC- and RNA-sequencing data analyses are described in the methods section. FDR < 0.05 was considered significant. **(A)** Multidimensional scaling plots showing the distance between all pairs of samples calculated using the root-mean-square of the log$_2$-fold changes of the top 500 most variable genes between any given two samples. **(B, C)** Volcano plot of −log$_{10}$(P-value) versus log$_2$ fold-change in RNA expression in *Kat6b*$^{-/-}$ versus *Kat6b*$^{+/+}$ NSPCs (B) and *Tg(Kat6b)* versus *Kat6b*$^{+/+}$ NSPCs (C). Significantly down-regulated genes shown in blue (B) or purple (C) and up-regulated genes in red (B) or orange (C) (FDR < 0.05). **(D)** Barcode plot showing the correlation between genes differentially expressed in *Kat6b*$^{-/-}$ versus *Kat6b*$^{+/+}$NSPCs and *Tg(Kat6b)* versus *Kat6b*$^{+/+}$ NSPCs. The block represents genes of the contrast *Kat6b*$^{-/-}$ (KO) versus *Kat6b*$^{+/+}$ (WT) NSPCs ordered from down-regulated, left, to up-regulated, right, with the t-statistic on the x axis. The vertical lines represent genes in the contrast *Tg(Kat6b)* versus

regulated in *Tg(Kat6b)* versus *Kat6b*[+/+] dorsal telencephalon were strongly associated with brain tissue expression (Fig 3D and E), whereas genes up-regulated in *Kat6b*[−/−] versus *Kat6b*[+/+] did not attract a specific tissue annotation and genes down-regulated in *Tg(Kat6b)* versus *Kat6b*[+/+] dorsal telencephalon were not significantly associated with expression in the brain (Fig S5I). As expected, a negative correlation was observed between the effects of loss and gain of KAT6B on gene expression (*P* = 0.02; Fig 3F). Genes down-regulated in *Kat6b*[−/−] versus *Kat6b*[+/+] E12.5 dorsal telencephalon were enriched for brain development and neuronal differentiation GO (BP) terms (Fig 3G; Table S3), whereas up-regulated genes did not prominently relate to brain development or function (Fig S5J). Loss of KAT6B resulted predominantly in the down-regulation of brain development genes (GO:0007420; FDR < 0.05; Fig 3H). Among the down-regulated genes were *Lhx2* (Hsu et al, 2015) and *Pou3f3* (Sugitani et al, 2002), genes individually required for neural precursor proliferation, and *Bhlhe22* and *Prdm8* (Ross et al, 2012), *Emx1* (Qiu et al, 1996), *Eomes* (Sessa et al, 2008), *Lhx2* (Porter et al, 1997), *Neurod2* (Olson et al, 2001), *Pou3f3* (Sugitani et al, 2002), *Otx1* (Acampora et al, 1996), and other genes that are essential for normal brain and neuronal development. Genes associated with central nervous system neuron development (GO:0021954) were highly enriched among down-regulated genes in the absence of KAT6B and enriched among up-regulated genes when *Kat6b* was overexpressed (*P* = 0.0008 and 0.005, respectively; Fig 3I and J). Gene expression levels correlated positively between E12.5 dorsal telencephalon and NSPCs and 16 genes were mutually down-regulated with transcriptome-wide significance in E12.5 dorsal telencephalon and NSPCs (Fig S5K–N). Comparably few genes were detected as differentially expressed in E15.5 *Kat6b*[−/−] versus *Kat6b*[+/+] or *Tg(Kat6b)* versus *Kat6b*[+/+] cortex (Fig S5H, O, and P; Table S3). This may reflect the increased cell type complexity of the E15.5 cortex compared with the E12.5 dorsal telencephalon or NSPCs. Overall, our data suggest that loss and gain of KAT6B predominantly affects the expression of brain development genes in NSPC and the developing cortex.

### Impaired proliferation, self-renewal, and neuronal differentiation in embryonic *Kat6b*[−/−] NSPCs

Embryonic NSPC colonies (neurospheres) derived from the *Kat6b*[−/−] E12.5 dorsal telencephalon appeared smaller, proliferated slower (*P* < 10[−6]) and gave rise to fewer secondary neurospheres compared with WT control NSPCs (*P* = 0.03; Fig 4A–C). *Kat6b*[−/−] NSPCs showed a 1.3-fold greater proportion of cells in $G_0$ of the cell cycle (*P* = 0.0003) and 8% decrease in the total percentage of proliferating (Ki67[+]) cells

(*P* = 0.03) compared with *Kat6b*[+/+] controls (Figs 4D–F and S6A and B). No difference in cell viability was observed between genotypes (Fig S6C–F).

Proliferating *Kat6b*[−/−] NSPCs contained a similar percentage of cells positive for the NSC marker and regulator SOX2 but showed a 30% reduction in the level of SOX2 protein levels per cell compared with *Kat6b*[+/+] cells (*P* = 0.02; Figs 4G and S6G–I). Under differentiating conditions, *Kat6b*[−/−] NSPCs gave rise to 46% fewer βIII tubulin[+] neurons (*P* = 0.01) and proportionally more GFAP[+] astrocytes (*P* = 0.01) than *Kat6b*[+/+] controls assessed by immunofluorescence (Fig 4H and I). Congruently, by flow cytometry, *Kat6b*[−/−] NSPCs gave rise to 68% fewer SOX2[−]βIII tubulin[+] neurons, 37% more SOX2[−]GFAP + astrocytes (*P* = 0.01 and *P* = 0.04, respectively) and also showed a 58% reduction in βIII tubulin protein levels (*P* = 0.009; Figs 4J–L and S7A–C). *Kat6b*[−/−] cortical neurons displayed shorter primary neurites (*P* = 0.003) and fewer secondary neurites (*P* = 0.04) compared with *Kat6b*[+/+] controls (Figs 4M and N and S7D–F).

Overall, loss of KAT6B appeared to affect neurogenesis from embryonic NSPCs at several levels including stem cell self-renewal, proliferation, neuronal differentiation, and neurite outgrowth. KAT6B appeared to be required for normal protein levels of the stem cell transcription factor SOX2 and the neuronal marker βIII tubulin.

### KAT6B promotes SOX family gene expression and directly targets the *Sox2* and *Pax6* genes

Among the brain development genes prominently affected by the absence of KAT6B was the SOX gene family. In NSPCs, SOX2 protein levels were reduced (Fig 4G), *Sox1, 2, 4, 8, 9,* and *21* were among the top 20 brain development genes and *Sox2, 5,* and *10* were among the top neural precursor proliferation genes that were down-regulated in *Kat6b*[−/−] versus *Kat6b*[+/+] NSPCs (Fig 2F and G). We therefore examined the effects of KAT6B on the SOX gene family in further detail. Loss of KAT6B prominently down-regulated most SOX family genes in NSPCs, E12.5 dorsal telencephalon, and E15.5 cortex (FDR within gene family = 0.05 to 3 × 10[−6]; Figs 5A–C and S8A). The SOX gene family was enriched among genes down-regulated in *Kat6b*[−/−] versus *Kat6b*[+/+] E12.5 dorsal telencephalon and E15.5 cortex (*P* = 0.003 and 0.001, respectively; Fig S8B and C). Congruently, some SOX genes were up-regulated when *Kat6b* was overexpressed (FDR within gene family = 0.05 to 3 × 10[−5]; Fig 5D–F). Consistent with KAT6B promoting *Sox2* expression, we observed a strong positive correlation between genes differentially expressed in *Kat6b*[−/−] versus control NSPCs and those lacking *Sox2* (Bertolini et al, 2019) (*P* = 0.0006; Figs 5G and S8D). Of 739 genes that were differentially

*Kat6b*[+/+] NSPCs, their height represents the log$_2$ fold-change. The worms indicate the enrichment, red for the genes up-regulated and blue for the genes down-regulated in *Tg(Kat6b)* versus *Kat6b*[+/+] NSPCs. **(E)** Top 20 gene ontology (GO) terms, biological processes (BP), enriched in genes down-regulated in *Kat6b*[−/−] versus *Kat6b*[+/+] NSPCs. **(F)** Log$_2$ fold-changes in RNA and in H3K9ac of the top 20 differentially expressed brain development genes (GO:0007420) in *Kat6b*[−/−] versus *Kat6b*[+/+] NSPCs. **(G)** Heatmap of neural precursor proliferation genes (GO:0061351) differentially expressed with FDR < 0.01 in *Kat6b*[−/−] versus *Kat6b*[+/+] NSPCs. Each column represents a NSCs isolate from and individual E12.5 embryo. **(H, I)** Correlation of read counts (log$_2$ CPM) levels between histone acetylation, DNA accessibility, POLR2A occupancy and RNA level in NSPCs for brain development genes (GO:0007420; (H)) and neural precursor proliferation genes (GO:0061351; (I)). **(J, K, L, M)** Correlation of read counts (log$_2$ RPKM) between H3K9ac in gene bodies (J, K, L, M) and RNA (J, L) or POLR2A occupancy at the TSS (K, M) assessing brain development genes (GO:0007420; (J, K)) or neural precursor proliferation genes (GO:0061351; (L, M)). **(N, O)** Correlation between log$_2$ fold changes in RNA and H3K9ac levels in gene bodies in *Kat6b*[−/−] versus *Kat6b*[+/+] NSPCs, assessing brain development genes (GO:0007420; (N)) or neural precursor proliferation genes (GO:0061351; (O)) down-regulated in *Kat6b*[−/−] versus *Kat6b*[+/+] NSPCs. Source data are available online for this figure.

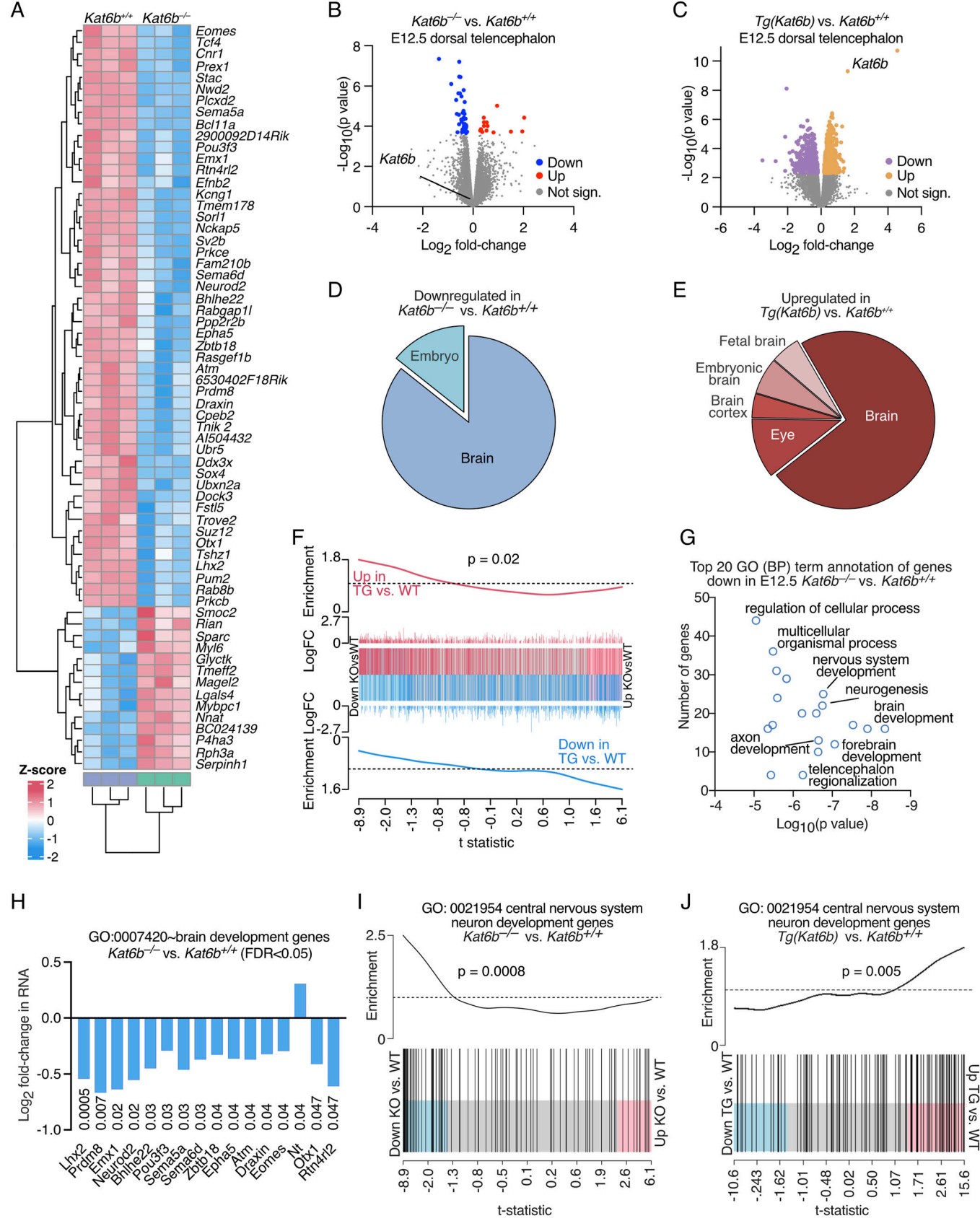

expressed in both, $Kat6b^{-/-}$ versus $Kat6b^{+/+}$ and $Sox2^{-/-}$ versus $Sox2^{+/+}$ NSPCs, 91% were changed in the same direction in both comparisons, 640 down-regulated and 31 up-regulated (Table S4), indicating similar effects of loss of KAT6B and loss of SOX2 on gene transcription in NSPCs. Of the top 20 genes down-regulates in either $Kat6b^{-/-}$ versus $Kat6b^{+/+}$ or $Sox2^{-/-}$ versus $Sox2^{+/+}$ NSPCs, 16 were down-regulated in both (Fig S8). Genes down-regulated in both comparisons were enriched for nervous system development GO terms (Fig S8F) and included genes mutated in human nervous system disorders, including autism spectrum disorder genes, epilepsy genes, developmental disorders, and several neurodegenerative disorders (Fig S8G).

To determine whether KAT6B could be a direct regulator of *Sox2*, we assessed if KAT6B bound the *Sox2* locus. Given the absence of reliable antibodies against KAT6B, we generated mice in which we tagged the endogenous KAT6B with a V5-tag ($Kat6b^{V5}$; Fig S8H and I). Unlike $Kat6b^{-/-}$ mice, which on a C57BL/6 genetic background die at birth, $Kat6b^{V5/V5}$ mice were viable and fertile, indicating that the KAT6B-V5 protein was functional. We used anti-V5 antibodies to detect the tagged form of KAT6B in $Kat6b^{V5/V5}$ mice by ChIP-qPCR. We observed a 3.2-fold enrichment of KAT6B-V5 binding to the *Sox2* promoter in $Kat6b^{V5/V5}$ compared with $Kat6b^{+/+}$ E15.5 cortex ($P$ = 0.002; Fig 5H). Furthermore, H3K9ac levels were reduced at the *Sox2* promoter in $Kat6b^{-/-}$ versus $Kat6b^{+/+}$ and increased in *Tg(Kat6b)* versus $Kat6b^{+/+}$ E15.5 cortex ($P$ = 0.01 and 0.006; Fig 5I and J). H3K14ac levels were increased in $Kat6b^{-/-}$ versus $Kat6b^{+/+}$ in E15.5 cortex ($P$ = 0.006; Fig 5K).

Apart from the *Sox* gene family, other neural development genes were also affected by loss or gain of KAT6B (Figs 2 and 3), including the paired homeodomain transcription factor gene *Pax6*, a NSC gene (Fig 2F and G). We found a 2-fold enrichment of KAT6B-V5 occupancy at the *Pax6* promoter in $Kat6b^{V5/V5}$ compared with $Kat6b^{+/+}$ E15.5 cortex ($P$ = 0.01; Fig 5L). H3K9ac levels were reduced ($P$ = 0.004) and H3K14ac levels increased ($P$ = 0.002) at the *Pax6* promoter in $Kat6b^{-/-}$ versus $Kat6b^{+/+}$ E15.5 cortex (Fig 5M and N). Acetylation at other histone residues was not affected by loss or gain of KAT6B at the *Sox2* and *Pax6* promoters (Fig S8J–M). Read depth plot examination suggested that loss of KAT6B was associated not only with a reduction in H3K9ac, but also a reduction in POLR2A occupancy at the *Sox2* and *Pax6* genes (Fig 5O) and at other *Sox* genes (Fig S9).

To further interrogate the effect of KAT6B on SOX2 expression, we assessed *Sox2* promoter activity in NSCs isolated from E12.5 dorsal telencephalon of $Kat6b^{+/+}$, $Kat6b^{+/-}$ and $Kat6b^{-/-}$ embryos that also expressed a *Gfp* reporter gene driven by the *Sox2* promoter

($Sox2^{GFP}$) (Arnold et al, 2011) by flow cytometry. Gating on SSEA1 and CD133 double positive cells, enriched for NSCs (Capela & Temple, 2002; Corti et al, 2007), we observed a gene-dose dependent reduction in *Sox2*-GFP levels in $Kat6b^{+/-}$ and $Kat6b^{-/-}$ compared with $Kat6b^{+/+}$ cells ($P$ = 0.04 and 0.0004; Fig 5P and Q), demonstrating a requirement for KAT6B for normal *Sox2* promoter activity.

Given the role of SOX2 as a driver of NSC multipotency and self-renewal (Suh et al, 2007; Miyagi et al, 2008; Favaro et al, 2009; Arnold et al, 2011), we postulated that SOX2 overexpression might rescue the proliferation defect of $Kat6b^{-/-}$ NSPCs (Fig 4A). Consistent with previous results (Fig 4A), $Kat6b^{-/-}$ NSPC expressing the empty vector ($Kat6b^{-/-}pMIG$) grew slower than $Kat6b^{+/+}pMIG$ NSPCs ($P$ = 2 × $10^{-6}$; Fig 5R). Remarkably, overexpression of *Sox2* in $Kat6b^{-/-}Sox2-pMIG$ NSPCs caused a partial rescue of proliferation. The proliferation capacity of $Kat6b^{-/-}Sox2-pMIG$ NSPCs was greater than $Kat6b^{-/-}pMIG^{+/+}$ NSPCs ($P$ = 0.01) and only marginally different from WT control cultures ($P$ = 0.05; Fig 5R). In addition, $Kat6b^{+/+}Sox2-pMIG$ NSPCs proliferated faster than $Kat6b^{-/-}Sox2-pMIG$ NSPCs ($P$ = 0.0002), suggesting that SOX2 drove fast NSPC proliferation in the presence of KAT6B and that SOX2 was unable to exert this strong proliferative effect to its full extent in the absence of KAT6B ($P < 10^{-6}$ for $Kat6b^{+/+}Sox2-pMIG$ vs. $Kat6b^{-/-}$ $Sox2-pMIG$ NSPCs).

## Discussion

In this study, we report the molecular effects of the histone acetyltransferase KAT6B during brain development and in embryonic NSPCs. We showed that KAT6B is essential for normal levels of H3K9ac at many genes, in particular brain development genes, including SOX family transcription factor genes. Notably, we show that KAT6B binds to and regulates the expression of a key regulator gene of neural precursor development, *Sox2*. In the absence of KAT6B, we observed a reduction in H3K9ac, *Sox2* promoter activity, *Sox2* mRNA, and SOX2 protein levels. We demonstrated direct binding of KAT6B to the *Sox2* promoter. Consistent with these results, *Sox2* overexpression partially rescued the proliferation defect of $Kat6b^{-/-}$ NSPCs in vitro.

Identification of KAT6B as promoting the expression of brain-specific genes in the developing cortex is consistent with the cognitive disorders caused by heterozygous mutations in the human *KAT6B* gene. The effects of KAT6B on the SOX gene family mirror the roles of its closely related paralogue KAT6A in promoting HOX, TBX and DLX family gene expression (Voss et al, 2009, 2012b; Sheikh

---

**Figure 3. Effects of loss of KAT6B on gene expression in the developing cortex.**
**(A, B, C, D, E, F, G, H, I, J)** RNA-sequencing data of E12.5 dorsal telencephalon. N = 3 $Kat6b^{-/-}$ versus 3 $Kat6b^{+/+}$ and 4 *Tg(Kat6b)* versus 4 $Kat6b^{+/+}$ embryos. RNA sequencing data were analyzed as described in the methods section. FDR < 0.05 was considered significant. **(A)** Heatmap of the genes differentially expressed in $Kat6b^{-/-}$ versus $Kat6b^{+/+}$ E12.5 dorsal telencephalon with transcriptome-wide significance (FDR < 0.05). **(B, C)** Volcano plot of -$\log_{10}$($P$ value) versus $\log_2$ fold-change in RNA expression in $Kat6b^{-/-}$ versus $Kat6b^{+/+}$ (B) and *Tg(Kat6b)* versus $Kat6b^{+/+}$ dorsal telencephalon (C). Down-regulated genes indicated in blue (B) or purple (C) and up-regulated genes in red (B) or orange (C) (FDR < 0.05). **(D, E)** Tissue annotation for genes down-regulated in $Kat6b^{-/-}$ versus $Kat6b^{+/+}$ (D) and up-regulated in *Tg(Kat6b)* versus $Kat6b^{+/+}$ dorsal telencephalon (E). **(F)** Barcode plot showing the correlation between genes differentially expressed in $Kat6b^{-/-}$ versus $Kat6b^{+/+}$ NSPCs and *Tg(Kat6b)* versus $Kat6b^{+/+}$ dorsal telencephalon. **(G)** Top 20 gene ontology (GO) terms, biological processes (BP), enriched in genes down-regulated in $Kat6b^{-/-}$ versus $Kat6b^{+/+}$ dorsal telencephalon. **(H)** Brain development genes (GO:0007420) differentially expressed in $Kat6b^{-/-}$ versus $Kat6b^{+/+}$ dorsal telencephalon. **(I, J)** Barcode plots depicting the results of gene set analyses for genes annotated with central nervous system development (GO:0021954) showing their enrichment in genes differentially expressed in $Kat6b^{-/-}$ versus $Kat6b^{+/+}$ NSPCs (I) and *Tg(Kat6b)* versus $Kat6b^{+/+}$ dorsal telencephalon (J).
Source data are available online for this figure.

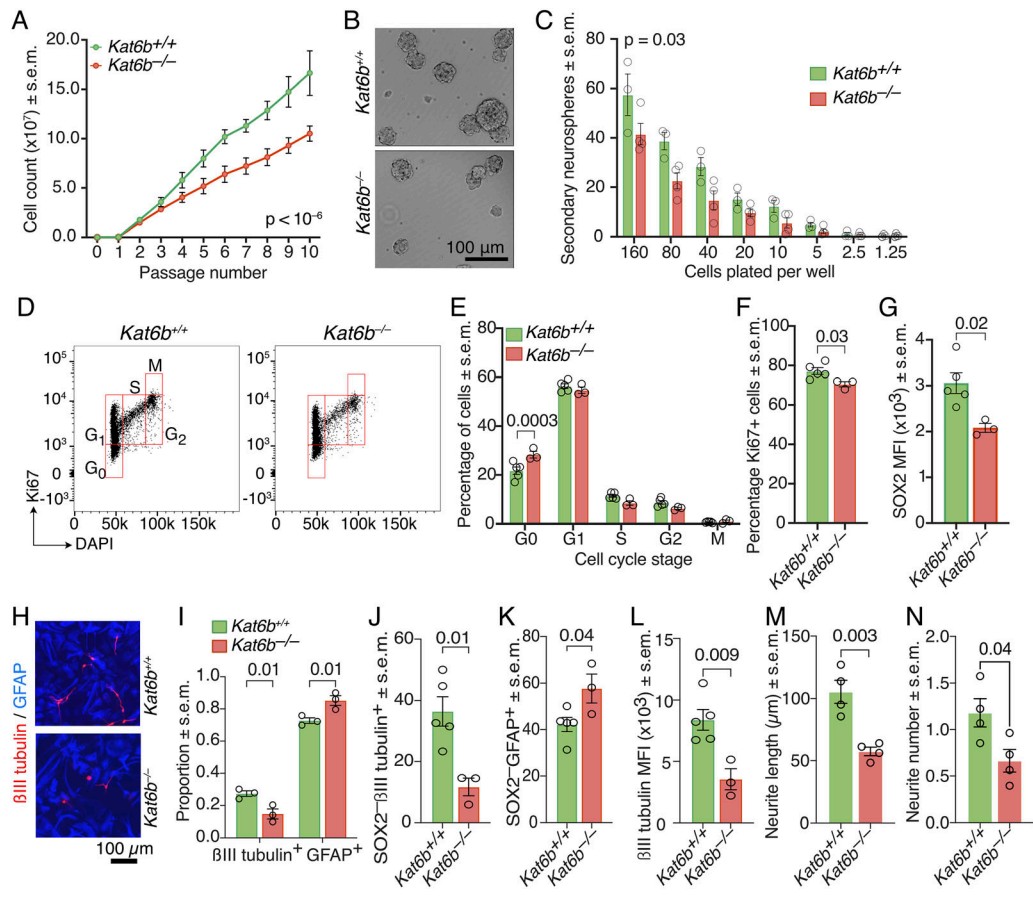

**Figure 4. Loss of KAT6B impairs embryonic NSPC proliferation, self-renewal, and neuronal differentiation.**
**(A)** Cumulative cell counts of NSPC cultures derived from E12.5 dorsal telencephalon tissue of $Kat6b^{+/+}$ or $Kat6b^{-/-}$ animals over 10 passages. **(B)** Representative images of $Kat6b^{+/+}$ and $Kat6b^{-/-}$ neurosphere colonies at passage 4. Scale bar 100 $\mu$m. **(C)** Number of secondary neurospheres generated by passage 5 $Kat6b^{+/+}$ and $Kat6b^{-/-}$ NSPC cultures in a dilution series. Circles represent averages of three technical replicates per three to four biological replicates per genotype and dilution. **(D, E, F)** Cell cycle analysis of $Kat6b^{+/+}$ and $Kat6b^{-/-}$ NSPCs. Representative flow cytometry plot of Ki67 versus DAPI staining of $Kat6b^{+/+}$ and $Kat6b^{-/-}$ NSPCs with the cell cycle stages $G_0$, $G_1$, S, $G_2$ and M indicated (D). **(E, F)** Percentage of passage 3 NSPCs in each cell cycle stage (E) and percentage of proliferating (Ki67$^+$) cells (F) in $Kat6b^{+/+}$ and $Kat6b^{-/-}$ NSPC cultures. **(G)** Median fluorescence intensity of SOX2 protein per cell detected by intranuclear immuno-staining and flow cytometry in passage 3 NSPCs. **(H, I)** Representative images (H) and quantification (I) of passage 3 $Kat6b^{+/+}$ and $Kat6b^{-/-}$ NSPC cultures grown for 6 d in differentiating conditions, stained to detect astrocytes (GFAP$^+$, blue) and neurons ($\beta$III tubulin$^+$, red). Scale bar 100 $\mu$m. **(J, K, L)** Quantification of the percentage of SOX2$^-\beta$III tubulin$^+$ (J) and SOX2$^-$GFAP$^+$ (K) cells in passage 3 $Kat6b^{+/+}$ and $Kat6b^{-/-}$ NSPCs grown for 6 d in differentiating conditions and median fluorescence intensity of $\beta$III tubulin per cell in $\beta$III tubulin$^+$ cells (L) assessed by flow cytometry. **(M, N)** Quantification of the average primary neurite length (M) and number of secondary neurites (N) in $\beta$III tubulin and DAPI stained $Kat6b^{+/+}$ and $Kat6b^{-/-}$ E16.5 cortical neurons after 5 d in culture. 100–200 neurons assessed per fetus. N = 3–4 (A, C), 3–5 (E, F, G, J, K, L), 3 (I) and 4 (M, N), embryos or fetuses per genotype. Each circle represents NSPCs (A, C, E, F, G, I, J, K, L) and E16.5 cortical neurons (M, N) derived from an individual embryo or fetus. Data are presented as mean ± SEM and were analyzed using a two-way ANOVA with Šidák correction (A, C, E, I) or unpaired $t$ test (F, G, J, K, L, M, N).
Source data are available online for this figure.

et al, 2015a; Vanyai et al, 2019). These findings suggest that activation of important developmental control gene families has been divided between this pair of closely related proteins, KAT6A and KAT6B. Unique among chromatin regulators, *Kat6b* is strongly regulated at the mRNA level in the forebrain. *Kat6b* mRNA levels are low before the onset of cortex development but between E11.5 and E15.5 they are strongly up-regulated during neural precursor proliferation and differentiation (Thomas et al, 2000).

Heterozygous null mutation in the human *KAT6B* gene causes a reduction in global histone H3 acetylation (Kraft et al, 2011). Our data suggest that the specific lysine target of KAT6B is H3K9 (here and Bergamasco et al [2024a, 2024b]). Loss of KAT6B did not abolish H3K9ac completely, indicating that other histone acetyltransferases

acetylate H3K9, too. This role has been ascribed to KAT2A (GCN5) and KAT2B (PCAF) (Jin et al, 2011) and to the KAT6B paralogue KAT6A (Voss et al, 2009, 2012b; Sheikh et al, 2015b; Vanyai et al, 2015, 2019; Lv et al, 2017; Yan et al, 2022). Interestingly, mRNA levels of the *Kat2a* gene were up-regulated in the absence of KAT6B and down-regulated when *Kat6b* was overexpressed, suggesting an attempt at compensatory up-regulation of this H3K9 acetyltransferase, which, however, did not achieve normal H3K9ac levels, gene expression or brain development. Similarly, the presence of *Kat6a* mRNA and the down-regulation of the histone deacetylase genes, *Hdac11* and *Sirt2*, were insufficient for a rescue of H3K9ac levels.

In addition, we found H3K23 to be a potentially cell type-specific acetylation target of KAT6B. H3K23 was reported to be a KAT6B target

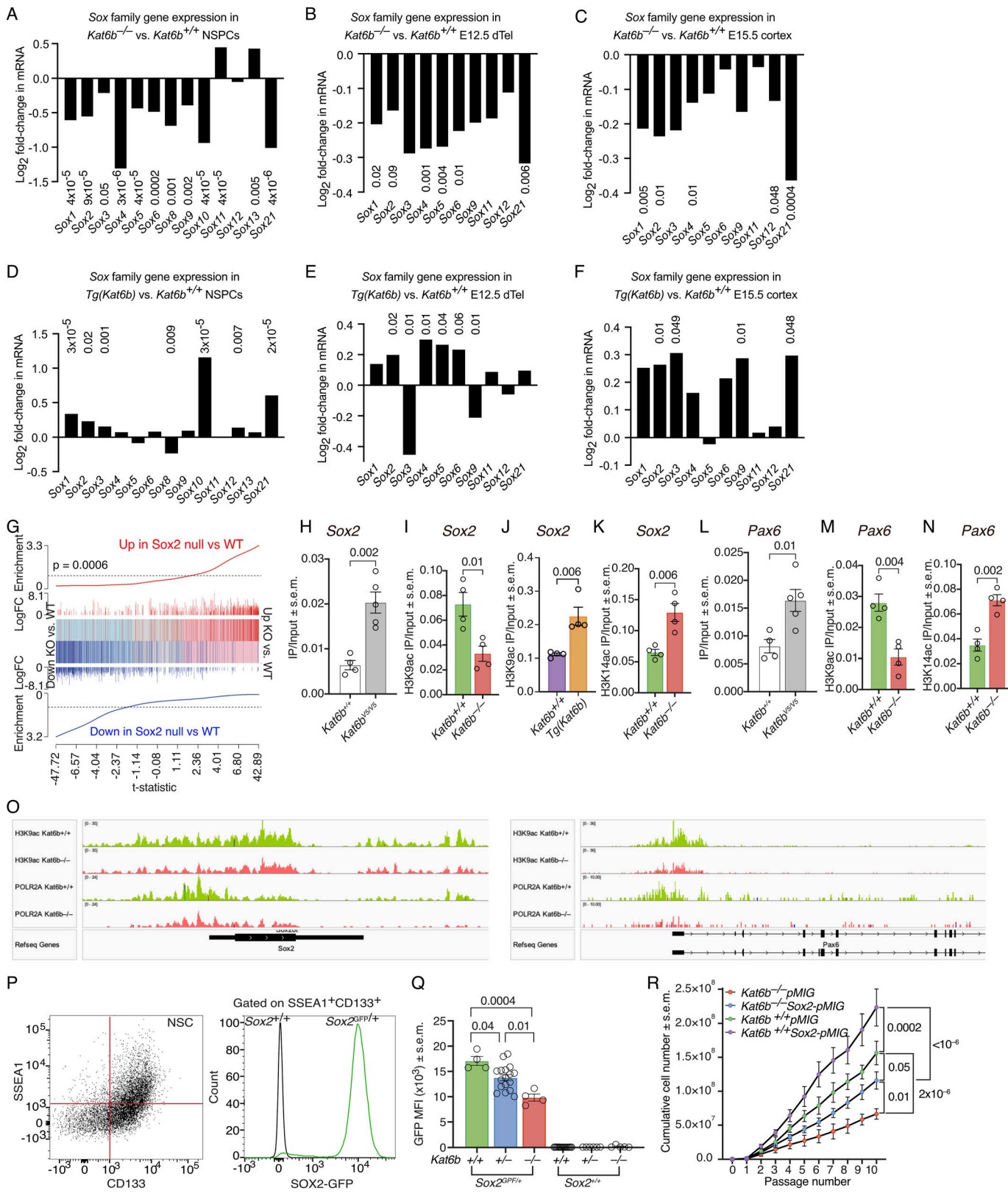

**Figure 5. KAT6B promotes expression of SOX family genes.**

**(A, B, C, D, E, F)** $Log_2$ fold change in mRNA for SOX family genes in $Kat6b^{-/-}$ versus $Kat6b^{+/+}$ (A, B, C) and $Tg(Kat6b)$ versus $Kat6b^{+/+}$(D, E) passage 5 NSPCs (A, D), E12.5 dorsal telencephalon (B, E) and E15.5 cortex (C, F). An average expression cut-off of > 32 counts per one million reads was used. FDRs < 0.05 shown above or below each bar. **(G)** Barcode plot showing the correlation of genes differentially expressed in $Kat6b^{-/-}$ versus $Kat6b^{+/+}$ NSPCs (this study) and $Sox2^{-/-}$ versus control NSPCs (Bertolini et al,

in previous studies (Simo-Riudalbas et al, 2015; Klein et al, 2019). H3K23ac is also catalyzed by KAT6A (Lv et al, 2017; Yan et al, 2020; Sharma et al, 2023). KAT6A is required for H3K9ac at specific gene loci in mouse embryonic fibroblasts (Sheikh et al, 2015b), in E10.5 embryos and tissue (Voss et al, 2009, 2012b; Vanyai et al, 2015, 2019) and in AML cells (Yan et al, 2022), as well as for H3K23ac in glioblastoma cells (Lv et al, 2017) and breast cancer cells (Sharma et al, 2023). These findings indicate that the histone lysine target of KAT6B and KAT6A may be cell-type dependent. Alternatively, KAT6A may be better able to compensate for a reduction in H3K23ac caused by loss of KAT6B in NSPCs in vitro than in the developing cortex in vivo.

Surprisingly, we also observed an increase in H3K14ac in E12.5 $Kat6b^{-/-}$ embryos and NSPCs along with a genome-wide increase in DNA accessibility. H3K14ac is dependent on another MYST family histone acetyltransferase, KAT7, in tissues and cell types during mouse development and in several human cell types (Kueh et al, 2011, 2020, 2023; Mishima et al, 2011; Feng et al, 2016). Given that histone H3 lysine 9 and 14 residues are only five amino acids apart, steric hindrance may occur between the KAT7 and the KAT6B multiprotein complexes. Reduced KAT6B complex occupancy at H3K9 in $Kat6b^{-/-}$ samples may provide an opportunity for increased access by the KAT7 complex. Alternatively, or in addition, KAT7, KAT6A, and KAT6B share protein complex members (Doyon et al, 2006) and therefore loss of KAT6B may increase protein complex partner availability for KAT7 and/or KAT6A complexes, altering the stoichiometric balance of MYST family histone acetyltransferase complexes within the cell. Notably, the increase in H3K14ac did not rescue the effects of loss of KAT6B and reduction in H3K9ac at the *Sox2* and *Pax6* gene and reduced expression of over 3,000 genes, indicating the importance of KAT6B and normal H3K9ac levels for neural development and suggesting that a reduction in H3K9ac cannot be compensated for by an increase in H3K14ac.

KAT6B and SOX2 appear to form a positive feedback loop. Not only does KAT6B bind the *Sox2* gene and promote *Sox2* expression (shown here), it has also been demonstrated that SOX2 binds the *Kat6b* gene in ES cells (Whyte et al, 2013; Cosentino et al, 2019) and NSCs (Bertolini et al, 2019). Positive feedback loops involving SOX2 have been shown to regulate NSPC self-renewal (SOX2 and EGFR [Hu et al, 2010]) and inhibit premature differentiation (SOX2 and SOX6 [Lee et al, 2014]). Similarly, a positive feedback loop may also exist between KAT6B and PAX6, whereby KAT6B binds to and activates *Pax6* gene expression (shown here), and PAX6 binds the *KAT6B* gene is ES cell-derived neural progenitors (Thakurela et al, 2016).

Consistent with SOX2 being a KAT6B gene target in the developing cerebral cortex, the *Kat6b* and *Sox2* genes are both expressed strongly in the dorsal telencephalon at E10.5 to E12.5 (Thomas et al, 2000; Thomas & Voss, 2004; Iwafuchi-Doi et al, 2011; Roberts et al, 2014) and in the developing cortex at E14.5–E17.5 (Thomas et al, 2000; Diez-Roux et al, 2011; Cánovas et al, 2015). Both genes are more strongly expressed in the developing cortex than in neighboring tissues. Importantly, there is a notable overlap in the phenotypic consequences of SOX2 and KAT6B deficiency in neural cell types. While the complete loss of SOX2 and KAT6B cause dissimilar phenotypes, namely very early embryonic lethality before the egg cylinder stage (Avilion et al, 2003) versus death at birth (Thomas et al, 2000), *Sox2* mRNA levels are not reduced to zero in the absence of KAT6B and therefore the homozygous null mutation of *Sox2* is not a relevant comparison in this context. In contrast, hypomorphic alleles of *Sox2* and *Kat6b* cause similar phenotypic anomalies. Mice carrying a *Sox2* hypomorphic allele, lacking 70–75% of endogenous mRNA levels, have reduced GABAergic neurons in the newborn cortex and adult olfactory bulb (Cavallaro et al, 2008) and impaired neurogenesis in the adult hippocampus and subventricular zone niches (Ferri et al, 2004). Similarly, $Kat6b^{gt/gt}$ adult mice have fewer GAD67+ GABAergic neurons (Thomas et al, 2000) and fewer neuroblasts in the rostral migratory stream (Merson et al, 2006). Neurons derived from *Sox2* deficient neuronal progenitors show impaired arborization and only weakly express markers of mature neurons (Cavallaro et al, 2008), consistent with impaired neurite outgrowth in $Kat6b^{-/-}$ E16.5 cortical neurons and reduced levels ßIII-tubulin protein in $Kat6b^{-/-}$ neurons shown here.

While our functional assessment focused on SOX2 as a key KAT6B gene target, other SOX family genes down-regulated in $Kat6b^{-/-}$ and up-regulated in *Tg(Kat6b)* samples also have important roles in brain development and NSC function. For example, *Sox1* is expressed during neural plate formation and its overexpression promotes neuronal differentiation in P19 cells (Pevny et al, 1998) and mouse NSPCs (Kan et al, 2004). SOX3 inhibits premature astrocyte differentiation by repressing gene targets of SOX9 in the developing mouse spinal cord (Klum et al, 2018) and SOX21 promotes neurogenesis in mouse hippocampal neurons (Matsuda et al, 2012). While it is tempting to attribute each defect in $Kat6b^{-/-}$ NSPCs to an individual gene target or a small collection of genes, it is more likely that the molecular consequences of loss of KAT6B described in this study result from the cumulative disruption of multiple direct and indirect gene targets. To this point, we only see a partial rescue of the proliferative defect of $Kat6b^{-/-}$ NSPCs

2019). **(H, I, J, K, L, M, N)** ChIP-qPCR results detecting: (H, L) KAT6B-V5 occupancy at the *Sox2* and *Pax6* promoters in in $Kat6b^{V5/V5}$ versus $Kat6b^{+/+}$ E15.5 cortex. **(I, J, M)** H3K9ac levels in $Kat6b^{-/-}$ versus $Kat6b^{+/+}$ (I, M) and *Tg(Kat6b)* versus $Kat6b^{+/+}$ E15.5 cortex (J) at the *Sox2* (I, J) and *Pax6* promoters (M). **(K, N)** H3K14ac levels in $Kat6b^{-/-}$ versus $Kat6b^{+/+}$ E15.5 cortex at the *Sox2* (K) and *Pax6* promoters (N). **(O)** Read depth plots of CUT&Tag sequencing reads displaying H3K9ac and POLR2A occupancy at the *Sox2* and *Pax6* genes showing representative plots of N = 4 $Kat6b^{+/+}$ and 4 $Kat6b^{-/-}$ NSPC isolates. **(P)** Representative flow cytometry plot showing the gating strategy for SSEA1 and CD133 double positive NSCs and histogram showing *Sox2*-GFP positive cells in $Sox2^{Gfp/+}$ and GFP negative $Sox^{+/+}$ cells isolated from E12.5 dorsal telencephalon tissue. **(Q)** Median fluorescence intensity of GFP in SSEA1⁺CD133⁺ cells from $Kat6b^{+/+}$, $Kat6b^{+/-}$ and $Kat6b^{-/-}$ embryos with and without the $Sox2^{Gfp}$ allele. **(R)** Cumulative growth curve for NSPCs isolated from $Kat6b^{-/-}$ and $Kat6b^{+/+}$ embryos and transfected with a *Sox2* overexpression vector (*Sox2-pMIG; pMSCV-Sox2-IRES-GFP II*) or empty vector (*pMIG; pMIG II*) at passage 3. N = 2–6 NSPC cultures, 3–4 E12.5 dorsal telencephalon or 4–5 E15.5 developing cortex samples per genotype (A, B, C, D, E, F, G), 4–5 E15.5 cortex samples (H, I, J, K, L, M, N), NSPCs from four embryos per genotype (O, R) and 4–17 E12.5 dorsal telencephalon samples (Q). Each circle represents NSPCs or tissue from an individual embryo or fetus (H, I, J, K, L, M, N, Q). Data were analyzed as stated in the methods section (A, B, C, D, E, F, G), using an unpaired *t* test (H, I, J, K, L, M, N), one-way (Q) or two-way ANOVA with Dunnett correction (R).

Source data are available online for this figure.

when *Sox2* is overexpressed, indicating that the reduction in SOX2 is not responsible for this phenotype alone.

In conclusion, we have provided a characterization of the effects of loss of KAT6B on histone acetylation and gene expression in general and in particular on the SOX gene family. We identified *Sox2* as a direct target of KAT6B, where KAT6B binds to the promoter of the *Sox2* gene, promotes H3K9ac and *Sox2* promoter activity, *Sox2* mRNA levels, and ultimately SOX2 protein levels. Finally, we showed that overexpression of *Sox2* can partially rescue a *Kat6b* loss of function phenotype, confirming a functional relationship between KAT6B, SOX2, and downstream events. Thus, our molecular analyses of KAT6B function reveal its importance for transcriptionally orchestrating neural development through H3K9ac and chromatin-based activation of development regulatory genes including the SOX2-driven molecular network.

# Materials and Methods

### Ethics statement

Animal experiments were conducted with approval of the WEHI Animal Ethics Committee and performed according to the Australian code for the care and use of animals for scientific purposes.

### Mouse husbandry

Mice were housed four to six animals in ventilated cages (AirLaw) and provided with γ-irradiated feed (Ridley AgriProducts; Barastoc) and sterilized water. Mice were kept in a 14 h light/10 h dark cycle. Noon of the day a vaginal copulation plug was first observed was defined as embryonic day 0.5 (E0.5).

### Mouse strains

We used *Kat6b*⁻ mice lacking exons 2–12 of the *Kat6b* gene, which we reported previously (Bergamasco et al, 2024a, 2024b), and *Tg(Kat6b)* BAC transgenic mice, which overexpress *Kat6b* ~4.5-fold, also reported previously (Bergamasco et al, 2024a). *Sox2-GFP* mice (B6; 129S-Sox2tm2Hoch/J [Arnold et al, 2011]) were obtained from the Jackson Laboratory. Mice were genotyped by PCR using primers displayed in Table S5.

### Generation of *Kat6b^V5* mice

A FLAG-V5-biotinylation sequence triple tag with a neomycin phosphotransferase cassette (Soler et al, 2010; Schwickert et al, 2014) was inserted by homologous recombination in *Escherichia coli* into the endogenous *Kat6b* gene after the last codon of KAT6B in a *Kat6b*-containing BAC, followed by targeting of the *Kat6b* locus in embryonic stem cells and germline chimera production to generate *Kat6b^Fag-V5-BIO* mice (*Kat6b^V5*). *Kat6b^V5* mice were crossed to *Cre-deleter* mice (Schwenk et al, 1995) to remove the neomycin phosphotransferase cassette. *Kat6b^V5* mice were genotyped using primers in Table S5.

### NSPC culture

NSPCs were derived from E12.5 dorsal telencephalon and cultured as free-floating neurosphere colonies as described (Rietze et al, 2001) in neurosphere medium (DMEM/F12 [12500-062; Gibco], 5 mM HEPES [H-4034; Sigma-Aldrich], 13.4 mM NaHCO$_3$ [G-7021; Sigma-Aldrich], 100 U/ml penicillin–streptomycin [15140-122; Gibco], 25 $\mu$g/ml insulin [I-6634; Sigma-Aldrich], 60 $\mu$M putrescine dihydrochloride [P-5780; Sigma-Aldrich], 100 $\mu$g/ml apo-transferrin [T-2252; Sigma-Aldrich], 30 nM selenium sodium salt [S-9133; Sigma-Aldrich], 20 nM progesterone [P-6149; Sigma-Aldrich], 0.2% BSA [A-3311; Sigma-Aldrich], 20 ng/ml EGF [78006.1; Neurocult] and 10 ng/ml FGF [78003; Neurocult]) at 37°C and 5% CO$_2$. Neurospheres were dissociated using Accutase (A1110501; Gibco) in a 37°C water bath for 4 min, counted using an automated cell counter (Countess; Thermo Fisher Scientific) and replated at 10,000 cells/cm$^2$. Neurospheres were imaged using an automated cell imager (ZOE, 1450031; Bio-Rad). Molecular and flow cytometry experiments were performed at passages 3–5. Secondary neurosphere formation assays were performed as described (Merson et al, 2006).

For NSPC differentiation, cells were dissociated using Accutase (A1110501; Gibco) and plated onto uncoated plates for flow cytometry or, for immunofluorescence, chamber slides (Nunc LabTek II, 154453; Thermo Fisher Scientific) pre-coated with 0.1 mg/ml poly-D-lysine (P6407; Sigma-Aldrich) followed by 10 $\mu$g/ml laminin (L2020; Sigma-Aldrich). Cells were grown in neurosphere medium, but without EGF or FGF and with 1% FCS for 6 d at 5% CO$_2$ and 37°C. For flow cytometry, cells were processed as described (Kueh et al, 2023). For immunofluorescence, cells were fixed in 4% PFA (wt/vol) for 20 min at RT, permeabilized and blocked in 10% (vol/vol) FCS and 0.03% (vol/vol) Triton X-100 for 1 h at RT. Slides were incubated O/N at 4°C with anti-βIII tubulin (1:500, G7121; Promega) and anti-GFAP (1:500, Z0334; Dako) in 10% FCS, washed (2 × 5 min PBS) and incubated with goat anti-mouse Alexa Fluor 546 (A11003; Invitrogen) and goat anti-rabbit AMCA (711-155-152; Jackson Immunoresearch) secondary antibodies at 1:400 for 1 h at RT in the dark. Slides were washed (2 × 5 min PBS) and mounted in Dako fluorescent mounting medium (S3023; Agilent).

### Flow cytometry

Intracellular flow cytometry analysis of NSPCs under proliferating and differentiating conditions was performed as described (Kueh et al, 2023) using antibodies listed in Table S6. For Ki67 versus DAPI cell cycle analysis, neurospheres were dissociated with Accutase (4 min, 37°C), fixed and permeabilized for 1 h on ice using the eBioscience Foxp3/Transcription factor staining buffer set (00-5523-00; Thermo Fisher Scientific), washed (2 × 5 min) in 1x permeabilization buffer from the same kit and resuspended in 100 $\mu$l/1 × 10$^6$ cells 2% FACS buffer (2% [vol/vol] FCS, 150 mM NaCl, 3.7 mM KCl, 2.5 mM CaCl$_2$·2H$_2$O, 1.2 mM MgSO$_4$·7H$_2$O, 0.8 mM K$_2$HPO$_4$, 1.2 mM KH$_2$PO$_4$, 11.5 mM HEPES, pH 7.4 in MQ-H$_2$O) containing 20 $\mu$l/1 × 10$^6$ cells anti-Ki67 FITC-conjugated antibody or isotype control (AB_396302; BD biosciences). Samples were incubated O/N at 4°C on a rocker, washed (2 × 5 min) in permeabilization buffer and resuspended in 100 $\mu$l/1 × 10$^6$ cells with 10 $\mu$l/ml DAPI (62248; Thermo Fisher Scientific). Cells were incubated on ice for 30 min,

washed (2 × 5 min) in permeabilization buffer and resuspended in 100 µl/1 × $10^6$ cells 2% FACS buffer.

For annexin V versus propidium iodine (PI) cell death analysis, neurospheres were dissociated with Accutase (4 min, 37°C) and processed using the Dead Cell Apoptosis Kit (V13242; Invitrogen) according to the manufacturer's instructions. For assessment of *Sox2*-GFP levels in E12.5 dorsal telencephalon, tissue was dissected and mechanically dissociated into a single cell suspension in 2% FACS buffer and passed through a 40 µm cell sieve (431750; Corning). Cells were centrifuged (200*g*, 5 min), supernatant removed, and cells resuspended in 2% FACS buffer containing anti-CD133-APC (17-1331-81; eBiosciences) and anti-SSEA1/CD15 (347420; BD Biosciences) antibodies at a 1:200 dilution. Samples were stained on ice for 1 h, washed (2 × 5 min) and resuspended in 2% FACS buffer containing 1:200 human anti-mouse PE-conjugated secondary antibody (made in house) and incubated on ice for 1 h. Samples were washed (2 × 5 min) and resuspended in 2% FACS buffer. Flow cytometry experiments were assessed on a LSRII or Fortessa flow cytometer (BD) at <7,500 events/sec and analyzed using FlowJo analysis software (version 10.7).

### E16.5 cortical neuron culture

E16.5 fetal cortices were dissected under a dissecting microscope (Zeiss). Isolated cortices were washed once in PBS (14190144; Gibco) before dissociation in 200 µl trypsin/EDTA (10006132; Sigma-Aldrich) for 10 min at 37°C. Excess trypsin was removed and replaced with 1 ml cortical neuron medium (DMEM/F12 [12500-062; Gibco], 5 mM HEPES [H-4034; Sigma-Aldrich], 13.4 mM NaHCO$_3$ [G-7021; Sigma-Aldrich], 100 U/ml penicillin–streptomycin [15140-122; Gibco], 25 µg/ml Insulin [I-6634; Sigma-Aldrich], 60 µM putrescine dihydrochloride [P-5780; Sigma-Aldrich], 100 µg/ml apo-transferrin [T-2252; Sigma-Aldrich], 30 nM selenium sodium salt [S-9133; Sigma-Aldrich], 20 nM progesterone [P-6149; Sigma-Aldrich], 0.2% BSA [A-3311; Sigma-Aldrich], and 1% FCS). Tissue was gently triturated, and cells passed through a 100 µm cell sieve (431751; Corning). Cells were plated onto chamber slides (C6932; Sigma-Aldrich) pre-coated with 0.1 mg/ml poly-D-lysine (P4832; Sigma-Aldrich) at 10,000 cells/cm$^2$, determined using an automated cell counter (Countess; Invitrogen). Cells were cultured at 37°C in 5% CO$_2$ for 5 d. On day 5, cells were fixed in 4% (wt/vol) paraformaldehyde (P6148; Sigma-Aldrich) in PBS, for 20 min at RT, nonspecific binding was blocked with 10% (vol/vol) FCS in PBS for 1 h at RT and cells incubated with anti-*β*III tubulin antibody (G7121; Promega) at 1:500 dilution in 10% FCS and 0.3% (vol/vol) Triton-X 100 in H$_2$O O/N at 4°C. Cells were washed (2 × 5 min) in PBS and incubated anti-mouse Alexa Fluro 546 antibody (A11003; Invitrogen) at 1:400 in 10% FCS for 1 h at RT. Cells were washed in PBS (2 × 5 min at RT) and incubated with 1 µg/ml DAPI (62248; Thermo Fisher Scientific) for 10 min at RT, washed in PBS (2 × 5 min) and mounted in Dako fluorescent mounting medium (S3023; Agilent).

### SOX2 overexpression in cultured NSPCs

The *Sox2* coding sequence (NM011443.4) was cloned by GeneArt Subcloning and Plasmid Services (Thermo Fisher Scientific) into a *pMSCV-IRES-GFP II* (*pMIG II*; Adgene [Holst et al, 2006]) vector using

Xho1/EcoR1 restriction enzyme sites to generate a *Sox2* expression construct (*Sox2-pMIG*; *pMSCV-Sox2-IRES-GFP II*). *Sox2-pMIG* or empty vector (*pMIG*; *pMIG II*) were virally packaged using human cells (Phoenix-AMPHO, ATCC CRL-3213). Phoenix cells were plated at a density of 3 × $10^6$ cells/10 cm plate in DMEM + 10% FCS for 24 h before transfection. 5.0 µg *Sox2-pMIG* or *pMIG* were combined with 250 µl 0.5 M CaCl$_2$, 250 µl MQ-H$_2$O, and 500 µl 2x HBSS (H4385; Sigma-Aldrich), vortexed for 10 s and incubated for 10 min at RT. The transfection mixture was added dropwise over Phoenix cells, swirling the plate to ensure even distribution. The following morning, transfection media was replaced with 6 ml complete neurosphere medium. Viral supernatant was collected at 24 and 48 h, passed through a 0.45 µm filter syringe (16533-K; Sartorius Australia) and added to freshly passaged NSPCs. Transfection efficiency was confirmed by detecting GFP using a fluorescent microscope (Zeiss).

### Western immunoblotting

Histones were extracted from whole E12.5 embryos, E12.5 dorsal telencephalon or cultured NSPCs by acid protein extraction. Cells or tissue were collected, washed in PBS (14190144; Gibco) containing 0.5 mM sodium butyrate (B5887; Sigma-Aldrich) and cOmplete EDTA-free protease inhibitor cocktail (11873580001; Roche) and collected by centrifugation (200*g*, 5 min). Samples were lysed in Histone acid lysis buffer (10 mM HEPES pH 7.9, 1.5 mM MgCl$_2$, 10 mM KCl, and 0.5 mM DTT) for 30 min at 4°C on a roller, collected by centrifugation (10,000*g*, 10 min), resuspended in 0.2 M H$_2$SO$_4$, incubated on ice for 1–2 h and dialyzed in dialysis tubing (Spectrum Spectra/Por Dialysis Membrane Tubing, molecular weight cut-off 20 kD; 08-607-067; Thermo Fisher Scientific) against 0.1 M acetic acid (A6283; Sigma-Aldrich) for 1 h at 4°C and MQ-H$_2$O overnight at 4°C. Protein concentrations were determined using a bicinchoninic acid assay (23225; Thermo Fisher Scientific). Acid extracted proteins were run on a 4–12% Bis–Tris gel (NP0322; Thermo Fisher Scientific) and transferred onto nitrocellulose membranes (926-31090; LI-COR Biosciences) for fluorescent detection or polyvinylidene fluoride membranes (45-3010040001; Kracheler scientific) for HPR detection. Membranes were blocked for 1 h at RT on a roller in blocking buffer (Intercept [PBS], 927-70001; LI-COR) for fluorescent detection or 5% skim milk in PBS + 0.1% (vol/vol) Tween-20 (P1379; Sigma-Aldrich) for HRP detection and probed with antibodies against acetylated histone lysines (Table S7) O/N at 4°C on a roller. Membranes were washed in PBS + 0.1% Tween-20 (P1379; Sigma-Aldrich) and incubated with secondary antibodies (Table S7) for 1 h at RT on a roller. For fluorescent detection, membranes were imaged and analyzed using an automated detection system (Odyssey, LI-COR). Acetylated histone H3 marks were normalized to pan H3 on the same membrane. For HRP detection, membranes were incubated with chemiluminescent HRP substrate (ECL, WBULS0500; Millipore) and exposed onto chemiluminescent film (GE healthcare). For detection of acetylated histone H4 lysine residues, membranes previously stained for an acetyl-histone H3 mark were incubated in 0.01% sodium azide (1 h RT), washed 3–4x in PBS and re-probed with an anti-acetyl H4 antibody. After exposure, membranes were washed in PBS and incubated with Ponceau S (A40000279; Thermo Fisher Scientific) for 5–10 min at RT. Exposed film and Ponceau S-stained

membranes were scanned and staining intensity quantified with Fiji image analysis software. Acetylated histone H3 and H4 lysine bands were normalized to an abundance basic protein band at 50 kD on Ponceau S-stained membranes.

## RNA isolation

Total RNA was collected from cultured NSPCs, E12.5 dorsal telencephalon or E15.5 cortex tissue using a kit (RNeasy mini kit, 74104; QIAGEN) according to the manufacturer's instructions and including the optional DNase I digest step. RNA quality and quantity were assessed on an automated electrophoresis system (Tapestation 4200, G2991BA; Agilent).

## RTqPCR

1 μg total RNA determined using a spectrophotometer (NanoDrop, Thermo Fisher Scientific) was used to generate cDNA using a cDNA synthesis kit (Superscript III, 18080051; Thermo Fisher Scientific) according to the manufacturer's instructions. cDNA was amplified using sequence-specific primers (Table S8) on an automated real time PCR system (LightCycle 480; Roche).

## Chromatin immunoprecipitation

Chromatin immunoprecipitating (ChIP) was performed as described (Voss et al, 2012a) using antibodies in Table S7. qPCR was performed using a LightCycle 480 instrument (Roche) and sequence-specific primers in Table S9.

## RNA sequencing and analysis

500 ng to 1 μg total RNA was used to generate sequencing libraries using a kit (TruSeq RNA prep kit v2, RS-122-2002; Illumina) according to the manufacturer's instructions. Samples were sequenced (NextSeq500; Illumina) to give 66 bp paired end reads.

## Analysis of E12.5 dorsal telencephalon and E15.5 cortex RNA sequencing data

All samples were aligned to the mm10 build of the mouse genome using Rsubread (v1.24.1) (Liao et al, 2019). In all cases, at least 85% of fragments were successfully mapped. Fragments overlapping genes were the summarized using Rsubread's featureCounts function. Genes were identified using Rsubread's inbuilt annotation for the mm10 genome. All sex-specific genes–*Xist* and those unique to the Y-chromosome – were removed to avoid sex biases. Genes with no official symbol were also discarded. Differential expression analyses were then carried out using the limma (v3.40.6) (Ritchie et al, 2015) and edgeR (v3.26.8) (Robinson et al, 2010) software packages. Independent analyses were performed for each mouse background–C57BL/6 ($Kat6b^{-/-}$ versus $Kat6b^{+/+}$) and FVB x BALB/c ($Tg(Kat6b)$ versus $Kat6b^{+/+}$).

For each dataset, expression-based filtering was first performed. For those samples from the C57BL/6 background, all genes that did not achieve a CPM greater than 1 in at least three samples were filtered. For those samples from the FVB x BALB/c background, all

genes that failed to achieve a CPM greater than 1 in at least four samples were removed. In both cases, sample composition was then normalized using the TMM method (Robinson & Oshlack, 2010).

Following filtering and normalization, the data in each analysis was transformed to $\log_2$-CPM. Differential expression between the genotype groups from each genetic background was then assessed using linear models and robust empirical Bayes moderated t-statistics with a trended prior variance (robust limma-trend pipeline) (Phipson et al, 2016). In the case of the C57BL/6 background samples, the linear models incorporated four surrogate variables to remove variation caused by a dissection error present in some samples as well as a litter batch effect. These were calculated using limma's wsva function. For the analysis of the FVB x BALB/c samples, a factor representing sample litter was included in the linear models, and sample weights were calculated using limma's array weights function with default parameters (Liu et al, 2015).

## Analysis of NSPC RNA sequencing data

An index combining the mm10 build of the mouse genome and *sacB* genomic sequence was first built using Rsubread's (v1.28.0) buildindex function. All samples were then aligned to this combined genome using Rsubread, achieving a mapping rate of at least 97% across all samples. Fragments overlapping genes were summarized using Rsubread's featureCounts. Genes were identified using Rsubread's inbuilt annotation to the mm10 genome. An additional line of annotation was added for the *sacB* sequence. Following fragment summarization, all genes with no symbol, ribosomal RNAs, non-protein coding immunoglobulin genes, predicted, unknown, and pseudo genes were all removed. Differential expression analyses were then carried out using limma (v3.40.6) and edgeR(v3.26.8). Independent analyses were conducted for each mouse genetic background - ($Kat6b^{-/-}$ versus $Kat6b^{+/+}$) and FVB x BALB/c ($Tg(Kat6b)$ versus $Kat6b^{+/+}$).

For each analysis, expression-based filtering was first performed. In each case, edgeR's filterByExpr function was used with default parameters. Following filtering TMM, normalization was applied to each data set.

The NSC from the C57BL/6 background was analyzed using a robust limma-voom pipeline. Thus, the data were first transformed to $\log_2$-CPM with associated precision weights using voom (Law et al, 2014). Differential expression between the genotype groups was then assessed using linear models and robust empirical Bayes moderated t-statistics.

The NSC from the FVB x BALB/c background was analyzed using a robust limma-voom with sample weights pipeline. Therefore, similarly to the C57BL/6 background analyses, the data was transformed to $\log_2$-CPM with associated precision weights using voom. Additional sample level weights were also calculated (limma voomWithQualityWeights function), and differential expression between the genotype groups was assessed using linear models and robust empirical Bayes moderated t-statistics.

For all analyses discussed here, the Benjamini and Hochberg method was applied to control the FDR below 5%. Pathway analyses were conducted using limma's goana and kegga functions. Multidimensional scaling plots, mean-difference (MD) plots, and

barcode plots were generated using limma's plotMDS, plotMD, and barcodeplot functions respectively. Heatmaps were generated using ComplexHeatmap.

## ATAC-sequencing and analysis

Using 50,000 NSPCs from $Kat6b^{+/+}$ or $Kat6b^{-/-}$ E12.5 dorsal telencephalon tissue, combined with 50,000 *Drosophila melanogaster* S2 cells spike-in (automated cell counter [Countess; Thermo Fisher Scientific Scientific]), ATAC-sequencing was performed as described (Buenrostro et al, 2015). Briefly, combined cells were lysed in 100 $\mu$l lysis buffer (10 mM Tris–HCl, pH 7.4, 10 mM NaCl, 3 mM MgCl2, 0.1% IGEPAL, and EDTA-free Complete protease inhibitors, made up in DNase-free $H_2O$). Nuclei were collected by centrifugation ($500g$, 10 min, 4°C) and resuspended in 50 $\mu$l TD buffer and TDE1 transposase (Nextera, #20034197; Illumina). Samples were digested (30 min at 37°C), purified using the QIAGEN MinElute PCR purification kit (28004; QIAGEN) and amplified using NEBNext High-Fidelity PCR master mix (M0541S) and P5/P7 indexing primer combinations (Table S10) under the following conditions: 72°C, 5 min, 98°C, 30 s, 10x cycles, 98°C, 10 s, 63°C, 30 s, 72°C, 1 min. Amplified samples were size-selected using 1.3x volumes AMPure beads (A63880; Beckman Coulter), eluted in 20 $\mu$l nuclease-free $H_2O$ and analyzed using a D1000 Tape and 4200 Tapestation (Agilent). Samples were processed on a high-throughput sequencing machine (NextSeq 2000; Illumina).

Following sequencing-read quality control, reads were aligned to the *Mus musculus* (mm39) and *D. melanogaster* (Dme1R6.32) genomes using Rsubread 2.12.3 (Liao et al, 2019). Mouse library sizes were normalized to *Drosophila* reads as controls, i.e., assuming that total *Drosophila* coverage should be equal across samples. Coverage was assessed across nonoverlapping 5 bp bins using the deepTools program (Ramírez et al, 2016). Read counts were obtained using the featureCounts function and the inbuilt mm39 annotation in Rsubread for gene promoters (TSS + 1 kb upstream) and transcription end sites (TES + 1 kb) of protein coding genes. Discontinued Entrez Gene IDs were excluded from analysis and genes with low counts were filtered using the filterByExpr function in edgeR. Counts were also obtained for active NSPC enhancers, defined as H3K4me1$^+$/H3K27ac$^+$, based on GSM2406793 and GSM2406791 (Bertolini et al, 2019). Enhancer regions overlapping with promoters or TES elements were removed. BEDTools (Quinlan & Hall, 2010) was used to identify overlapping H3K4me1/H3K27ac enriched regions. Differential accessibility upon gain or loss of *Kat6b* was assessed using quasi-likelihood generalized linear models in edgeR (Chen et al, 2016, 2024 *Preprint*).

## CUT&Tag sequencing and analysis

Cleavage Under Targets and Tagmentation (CUT&Tag) sequencing was performed using 50,000 NSPCs combined with 50,000 *D. melanogaster* S2 cells, as described in Kaya-Okur et al (2019), with minor modifications as described (Wichmann et al, 2022), using antibodies against H3K9ac, H3K14ac, H3K23ac, or RNA polymerase II, subunit A (POLR2A) (Table S7) and indexing primers (Table S10). Libraries were sequenced (NextSeq 2000; Illumina).

Reads were aligned to the *M. musculus* (mm39) and *D. melanogaster* (Dme1R6.32) genomes using Rsubread. Mouse library sizes were normalized using the *Drosophila* read content and normalized read abundance assessed at the following genomic regions.

(i) Gene promoters, defined as up to 1 kb upstream of the TSS to the TSS of protein coding genes.
(ii) Active NSPC enhancers, defined as H3K4me1$^+$H3K27ac$^+$, based on GSM2406793 and GSM2406791 (Bertolini et al, 2019).

Enhancer regions overlapping with a promoter or TES downstream regions were removed. Overlapping and nonoverlapping H3K4me1 and H3K27ac enriched regions were identified using BEDTools. The number of reads pairs overlapping each genomic region was summarized using Rsubread's featureCounts function (Liao et al, 2014). Coverages for non-overlapping 5 bp bins were computed using deepTools. Differential coverages upon gain or loss of *Kat6b* were assessed using quasi-likelihood generalized linear models in edgeR (Chen et al, 2016, 2024 *Preprint*).

## Statistics

The specific statistical tests employed and the number of biological replicates for each experiment is detailed in the figure legends. Statistical analyses were performed in Prism Graphpad Version 8.3.1 for Mac (GraphPad Software), except the analyses of the RNA-sequencing, ATAC-seq and CUT&Tag sequencing data, which are provided above in the methods sections.

# Data Availability

The RNA-sequencing, CUT&Tag sequencing, and ATAC-sequencing data are accessible at NCBI GEO under the accession numbers GSE280783 (NSPC RNA-seq data), GSE280784 (developing cortex RNA-seq data), GSE267672 (CUT&Tag data) and GSE267675 (ATAC-seq data).

# Supplementary Information

# Acknowledgements

The authors would like to thank R May, C Burström, and L Potenza for excellent technical support. S Wilcox for exceptional technical service. F Dabrowski, L Wilkins, N Blasch, S Bound, E Boyle, J Gilbert, S Oliver, and L Johnson for expert animal care. MI Bergamasco was supported by an Australian Government Postgraduate Award. The program of work was supported by the Pamela and Lorenzo Galli Charitable Trust and by the Australian National Health and Medical Research Council through Project Grants 1010851 to AK Voss and T Thomas; 1160517 to T Thomas; Ideas Grant 2010711 to T Thomas; Research Fellowships 1003435 to T Thomas, 575512 and 1081421 to AK Voss, and 1154970 to GK Smyth and Investigator Grants 1176789 to AK Voss and 2025645 to GK Smyth; through the Independent Research Institutes Infrastructure Support Scheme; by the Chan Zuckerberg Initiative

through grant 2021-237445 to GK Smyth and by the Victorian Government through an Operational Infrastructure Support Grant.

## Author Contributions

MI Bergamasco: formal analysis, validation, investigation, visualization, methodology, and writing—original draft, review, and editing.

W Abeysekera: formal analysis, visualization, methodology, and writing—original draft.

AL Garnham: formal analysis, visualization, methodology, and writing—original draft.

Y Hu: formal analysis.

CSN Li-Wai-Suen: formal analysis.

BN Sheikh: formal analysis, supervision, and methodology.

GK Smyth: formal analysis, supervision, funding acquisition, and methodology.

T Thomas: conceptualization, formal analysis, supervision, funding acquisition, methodology, and writing—original draft, review, and editing.

AK Voss: conceptualization, formal analysis, supervision, funding acquisition, visualization, methodology, project administration, and writing—original draft, review, and editing.

## Conflict of Interest Statement

The T Thomas and AK Voss laboratories have received research funding from the Cooperative Research Centre for Cancer Therapeutics (CRC-CTx).

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
