## [Reviewer comments · Life Science Alliance]

Life Science Alliance

KAT6B is required for histone 3 lysine 9 acetylation and SOX gene expression in the developing brain

Maria Bergamasco, Waruni Abeysekera, Alexandra Garnham, Yifang Hu, Connie Li-Wai-Suen, Bilal Sheikh, Gordon Smyth, Tim Thomas, and Anne Voss

DOI: <https://doi.org/10.26508/lsa.202402969>

Corresponding author(s): Anne Voss, Walter and Eliza Hall Institute of Medical Research

Review Timeline:	Submission Date:	2024-07-31
	Editorial Decision:	2024-07-31
	Revision Received:	2024-10-20
	Editorial Decision:	2024-10-21
	Revision Received:	2024-11-01
	Accepted:	2024-11-04

Transaction Report:

Please note that the manuscript was previously reviewed at another journal and the reports were taken into account in the decision-making process at Life Science Alliance. Since the original reviews are not subject to Life Science Alliance's transparent review process policy, the reports and author response cannot be published.

July 31, 2024

Re: Life Science Alliance manuscript #LSA-2024-02969-T

Dr. Anne K. Voss
Walter and Eliza Hall Institute of Medical Research
Development and Cancer Division
1G Royal Parade
Melbourne, Parkville, Victoria 3052
Australia

Dear Dr. Voss,

Thank you for submitting your manuscript entitled "KAT6B is required for histone 3 lysine 9 acetylation and SOX gene expression in the developing brain" to Life Science Alliance. We invite you to submit a revised manuscript addressing Reviewer 1's comments.

When submitting the revision, please include a letter addressing the reviewer's comments point by point.

Thank you for this interesting contribution to Life Science Alliance. We are looking forward to receiving your revised manuscript.

Sincerely,

Eric Sawey, PhD
Executive Editor
Life Science Alliance
<http://www.lsa-journal.org>

B. MANUSCRIPT ORGANIZATION AND FORMATTING:

October 21, 2024

RE: Life Science Alliance Manuscript #LSA-2024-02969-TR

Prof. Anne K. Voss
Walter and Eliza Hall Institute of Medical Research
Development and Cancer Division
1G Royal Parade
Melbourne, Parkville, Victoria 3052
Australia

Dear Dr. Voss,

Thank you for submitting your revised manuscript entitled "KAT6B is required for histone 3 lysine 9 acetylation and SOX gene expression in the developing brain". We would be happy to publish your paper in Life Science Alliance pending final revisions necessary to meet our formatting guidelines.

- please be sure that the authorship listing and order is correct
- please incorporate your supplemental material file into your main manuscript text; please upload your supplemental figures as single files and add the supplementary figure legends to the main figure legend section
- please make sure any table files are uploaded as editable doc or excel files or they're included in the doc file of your main manuscript text
- please include a Data Availability statement at the end of the Materials and Methods section that contains access information to the multiple sequencing datasets generated
- please add sizes next to all blots

LSA now encourages authors to provide a 30-60 second video where the study is briefly explained. We will use these videos on social media to promote the published paper and the presenting author (for examples, see <https://docs.google.com/document/d/1-UWCfbE4pGcDdcgzcmiuJI2XMBJnxKYeqRvLLrLS08s/edit?usp=sharing>). Corresponding or first-authors are welcome to submit the video. Please submit only one video per manuscript. The video can be emailed to contact@life-science-alliance.org

A. FINAL FILES:

B. MANUSCRIPT ORGANIZATION AND FORMATTING:

Sincerely,

November 4, 2024

RE: Life Science Alliance Manuscript #LSA-2024-02969-TRR

Prof. Anne K. Voss
Walter and Eliza Hall Institute of Medical Research
Development and Cancer Division
1G Royal Parade
Melbourne, Parkville, Victoria 3052
Australia

Dear Dr. Voss,

Thank you for submitting your Research Article entitled "KAT6B is required for histone 3 lysine 9 acetylation and SOX gene expression in the developing brain". It is a pleasure to let you know that your manuscript is now accepted for publication in Life Science Alliance. Congratulations on this interesting work.

DISTRIBUTION OF MATERIALS:

Again, congratulations on a very nice paper. I hope you found the review process to be constructive and are pleased with how the manuscript was handled editorially. We look forward to future exciting submissions from your lab.

Sincerely,
